# Paradoxical effects on voltage-gated Na⁺ conductance in adrenal chromaffin cells by twin vs single high intensity nanosecond electric pulses

**Lisha Yang**[1], **Sophia Pierce**[1], **Indira Chatterjee**[2], **Gale L. Craviso**[1], **Normand Leblanc**[1]*

**1** Department of Pharmacology, University of Nevada, Reno School of Medicine, Reno, NV, United States of America, **2** Department of Electrical and Biomedical Engineering, College of Engineering, University of Nevada, Reno, NV, United States of America

* nleblanc@unr.edu

**Data Availability Statement:** All relevant data are within the manuscript and its Supporting Information files.

## Abstract

We previously reported that a single 5 ns high intensity electric pulse (NEP) caused an E-field-dependent decrease in peak inward voltage-gated Na⁺ current ($I_{Na}$) in isolated bovine adrenal chromaffin cells. This study explored the effects of a pair of 5 ns pulses on $I_{Na}$ recorded in the same cell type, and how varying the E-field amplitude and interval between the pulses altered its response. Regardless of the E-field strength (5 to 10 MV/m), twin NEPs having interpulse intervals ≥ than 5 s caused the inhibition of TTX-sensitive $I_{Na}$ to approximately double relative to that produced by a single pulse. However, reducing the interval from 1 s to 10 ms between twin NEPs at E-fields of 5 and 8 MV/m but not 10 MV/m decreased the magnitude of the additive inhibitory effect by the second pulse in a pair on $I_{Na}$. The enhanced inhibitory effects of twin vs single NEPs on $I_{Na}$ were not due to a shift in the voltage-dependence of steady-state activation and inactivation but were associated with a reduction in maximal Na⁺ conductance. Paradoxically, reducing the interval between twin NEPs at 5 or 8 MV/m but not 10 MV/m led to a progressive interval-dependent recovery of $I_{Na}$, which after 9 min exceeded the level of $I_{Na}$ reached following the application of a single NEP. Disrupting lipid rafts by depleting membrane cholesterol with methyl-β-cyclodextrin enhanced the inhibitory effects of twin NEPs on $I_{Na}$ and ablated the progressive recovery of this current at short twin pulse intervals, suggesting a complete dissociation of the inhibitory effects of twin NEPs on this current from their ability to stimulate its recovery. Our results suggest that in contrast to a single NEP, twin NEPs may influence membrane lipid rafts in a manner that enhances the trafficking of newly synthesized and/or recycling of endocytosed voltage-gated Na⁺ channels, thereby pointing to novel means to regulate ion channels in excitable cells.

**Funding:** The work was supported by grants to GLC and NL from the Air Force Office of Scientific Research (AFOSR Grant # FA9550-14-1-0018 and FA9550-15-1-0517). AFOSR URL: https://www.wpafb.af.mil/afrl/afosr/. The funders had no role in study design, data collection and analysis, decision to publish, or preparation of the manuscript.

**Competing interests:** The authors have declared that no competing interests exist.

# Introduction

The advent of systems to deliver nanosecond duration electric pulses (NEPs) to biological cells has sparked a great deal of interest in exploring uses of these pulses for biomedical applications, such as neurostimulation. An example of the potential use of NEPs for neurostimulation is the report by Jiang and Cooper [1] that a single 12 ns pulse was capable of activating skin nociceptors. Also, a recent study by Casciola *et al.* [2] demonstrated that repeated stimulation of isolated peripheral nerves of *Xenopus leavis* by 12 ns pulses continuously elicited action potentials without damaging the nerve fibers. Previous studies from our group additionally have shown that a 5 ns pulse can stimulate catecholamine release in neuroendocrine adrenal chromaffin cells by causing $Ca^{2+}$ influx via voltage-gated calcium channels (VGCCs) [3].

Efforts aimed at elucidating the mechanisms by which NEPs stimulate neural cells and tissues have revealed some interesting differences with respect to the involvement of voltage-gated $Na^+$ channels. In peripheral nerve, 12 ns pulse exposure triggers $Na^+$ influx via voltage-gated $Na^+$ channels, which is responsible for the generation of action potentials [2]. In bovine chromaffin cells, in contrast, $Na^+$ influx via voltage-gated $Na^+$ channels is not responsible for the membrane depolarization that evokes VGCC activation in cells exposed to a 5 ns pulse [4]. Instead, membrane depolarization appears to be the result of $Na^+$ influx via putative nanopores [4, 5]. Moreover, a 5 ns pulse actually causes an inhibition of voltage-gated $Na^+$ channels in these cells [6].

Nesin *et al.* [7, 8] were the first to report that voltage-gated $Na^+$ channels can be inhibited by NEPs. In their studies, a single 300 ns pulse was found to cause a prolonged inhibition of $Na^+$ conductance in several excitable cell types that included NG108 cells, GH3 cells, as well as adrenal chromaffin cells. The inhibitory effect was enhanced when the E-field magnitude of the applied pulse was increased. Of interest to us was whether the much shorter duration 5 ns pulses that we had been studying also caused inhibitory effects on voltage-gated $Na^+$ channels in chromaffin cells and if so, how the inhibition was affected by E-field intensity. To this end we recently evaluated the effects of a single 5 ns pulse on fast voltage-dependent $Na^+$ currents ($I_{Na}$) recorded in whole-cell voltage clamped bovine chromaffin cells. The gating properties of the net current were consistent with those of tetrodotoxin-sensitive $Na^+$ channels, and of $Na_V1.7$, the pore-forming $Na^+$ channel subunit thought to be the main voltage-gated $Na^+$ channel expressed in bovine chromaffin cells [9–12]. Our results showed that a single 5 ns pulse applied at 5 MV/m, an E-field magnitude that is just above the threshold for evoking responses in these cells [13] produced a rapid inhibition (~ 4%) of $I_{Na}$ at a test potential of +10 mV [6], which was in agreement with the inhibition of voltage-gated $Na^+$ channels previously reported in these cells [8]. The decrease in $I_{Na}$ was associated with a decrease in maximal conductance without a change in its voltage-dependence of steady-state activation and inactivation [6]. Our results also showed that a single 5 ns pulse at the higher E-field of 8 MV/m and 10 MV/m produced a rapid inhibitory effect that was ~ twofold greater (from ~ 4% to ~ 9%). The purpose of this study was to determine whether the application of a second pulse of identical characteristics leads to additive effects on $I_{Na}$.

Our data indicate that the application of a second NEP produced as anticipated an additive inhibitory effect on $I_{Na}$ but surprisingly, only when the interval between the pulses exceeded 100 ms. In addition, the second pulse in a pair caused a paradoxical time-dependent recovery of the $Na^+$ current after inhibition, a phenomenon that varied with E-field magnitude and the time interval between the pulses, and that appeared to involve lipid rafts. Our study strengthens the notion that voltage-dependent ionic currents can be altered by NEPs and offers a new perspective on how the application of multiple pulses could be tailored to augment or attenuate inhibitory effects on voltage-gated $Na^+$ channel activity in excitable cells.

## Materials and methods

### Chromaffin cell culture

As described previously [14], chromaffin cells were isolated by collagenase digestion of the medulla from fresh bovine adrenal glands obtained from a local abattoir (Wolf Pack Meats, University of Nevada, Reno). After isolation, the cells were maintained in suspension culture in Petri dishes at 36.5˚C under a humidified atmosphere of 5% $CO_2$ in Ham's F-12 medium supplemented with 10% bovine calf serum, 1% antibiotic (100 U/ml penicillin, 100 μg/ml streptomycin, 0.25 μg/ml fungizone) and 6 μg/ml cytosine arabinoside. Cells were used up until two weeks in culture. For experiments, single, isolated cells were obtained by treatment of cell aggregates with the protease dispase [15] and attached to fibronectin-coated glass coverslips.

### Whole-cell patch clamp experiments

Coverslips containing the attached cells were placed in a perfusion chamber positioned on the stage of a Nikon Eclipse TS100 inverted microscope. The cells were continuously perfused at a rate of 0.5 ml/min with a balanced salt solution (BSS) consisting of 145 mM NaCl, 5 mM KCl, 2 mM $CaCl_2$, 1.2 mM $Na_2HPO_4$, 1.3 mM $MgCl_2$, 10 mM glucose and 15 mM Hepes, pH 7.4 (adjusted with NaOH). The bath solution was connected to the ground via an Ag/AgCl pellet. The patch pipette internal solution contained 10 mM NaCl, 30 mM KCl, 110 mM K-gluconate, 1 mM $MgCl_2$, 10 mM EGTA, 3 mM Mg-ATP, and 10 mM Hepes, pH 7.2 (adjusted with KOH). Whole-cell currents were recorded in voltage clamp mode using an Axopatch 200B amplifier and Digidata 1322A (Axon Instruments, Sunnyvale, CA) or Digidata 1550B data acquisition system (Molecular Devices, Sunnyvale, CA), and PClamp software (version 8.2 or 11, Molecular Devices, Sunnyvale, CA) at a sampling rate of 20 kHz and low-pass filtering at 1 kHz. The series resistance ($R_s$), which was compensated to 90%, varied between 7 and 32 MΩ. The seal resistance ($R_{seal}$) varied between 1.1 and 3.0 GΩ and was measured by applying a +10 mV test pulse from a holding potential of –70 mV. Cell capacitance ($C_m$) ranged between 4 and 12 pF. All recordings were performed at ambient room temperature. Fire-polished patch micropipettes having a tip size of 0.8–1.2 μm with a typical resistance of ~ 3 MΩ were fabricated from borosilicate glass (#BF150-110-7.5, Sutter Instruments, Novato, CA) using a P-97 pipette puller (Sutter Instruments, Novato, CA) and fire-polished with a MF-830 microforge (Narishige, Tokyo, Japan). The cell being recorded was viewed with a 40X objective and a bright field image of the cell was captured with a CoolSnap HQ DIFF CCD camera (Photometrics, Tucson, AZ) and SimplePCI software (version 6.6.0.0, Hamamatsu Corporation, Hamamatsu City, Japan) at the beginning and end of each experiment. For recordings obtained in the absence of extracellular $Na^+$, an equimolar concentration of N-methyl-D-glucamine ($NMDG^+$) was used to replace $Na^+$ in the external bath solution. For recordings conducted under $K^+$-free conditions, the patch pipette internal solution contained 10 mM NaCl, 20 mM tetraethylammonium chloride (TEA-Cl), 10 mM CsCl, 110 mM aspartic acid, 1 mM $MgCl_2$, 10 mM EGTA, 3 mM Mg-ATP, and 10 mM Hepes, pH 7.2 (adjusted with CsOH) at room temperature. The external bath solution consisted of 145 mM NaCl, 10 mM TEA-Cl, 1.2 mM $Na_2HPO_4$, 2 mM $CaCl_2$, 1.3 mM $MgCl_2$, 5.5 mM glucose and 10 mM Hepes, pH 7.4 (adjusted with CsOH). For the $Ca^{2+}$-free experiments, an equimolar concentration of $MgCl_2$ was used to replace $CaCl_2$ in the external bath solution.

The effects of NEPs on peak inward currents were recorded by applying a total of 200 voltage clamp steps, with a 3 s interval between each step as described in our previous work [6]. The voltage clamp protocol consisted of applying a 100 ms voltage step to +10 mV from a

holding potential of –70 mV. A single pulse or a pulse pair was applied 1.5 s after the 21$^{st}$ voltage clamp step. The detailed protocols are described in the supplemental information (S1 Fig). Currents of the 8 voltage steps prior to NEP exposure were averaged and used as baseline.

The NEP-induced leak current ($I_{leak}$) was recorded during a 2 s voltage ramp protocol ranging from –70 to +80 mV from a holding potential of –70 mV. The ramp protocol was applied 5 times prior to the application of a pair of NEPs to determine control baseline membrane current, and 15 times following the application of twin pulses. The interval between each ramp was 6 s. $I_{leak}$ was measured in K$^+$-free pipette internal solution and in K$^+$-free external bath solution containing 20 μM nitrendipine. $I_{leak}$ was determined by subtracting the ramp current measured after the pair of NEPs from the baseline ramp current.

Current-voltage (I-V) relationships for inward currents were generated using a voltage step protocol consisting of 50 ms steps ranging from –70 mV to +80 mV applied in 10 mV increments every 2 s. Steady-state activation curves for Na$^+$ currents were constructed by measuring the peak Na$^+$ conductance ($G_{Na}$) calculated from the equation:

$$G_{Na} = \frac{I_{Na}}{V - E_{rev}},$$

where $I_{Na}$ is the peak Na$^+$ current during the test depolarization (V), and $E_{rev}$ is the reversal potential of the inward current determined as the voltage producing null current on the I-V relationship. Data were normalized to maximum peak conductance ($G_{max}$) and fitted to a Boltzmann function:

$$\frac{G_{Na}}{G_{max}} = b + \frac{a}{1 + e^{(\frac{V0.5 - V}{k})}},$$

from which the slope factor ($k$) and half-maximal activation voltage ($V_{0.5}$) were derived.

The steady-state inactivation curve of $I_{Na}$ was generated by holding the cell at –70 mV and applying a series of 1 s conditioning voltage clamp steps ranging from –100 to 0 mV in 10 mV increments, with each voltage step followed by a constant 100 ms test pulse to +10 mV to record $I_{Na}$. $I_{Na}$ was normalized to its respective maximal value and plotted as a function of the conditioning potential. Normalized peak inward currents were plotted as a function of the conditioning potentials and the data fitted to an appropriate Boltzmann function:

$$\frac{I_{Na}}{I_{max}} = b + \frac{a}{1 + e^{(\frac{V - V0.5}{k})}},$$

from which the slope factor ($k$) and half-maximal inactivation voltage ($V_{0.5}$) were derived.

## NEP exposure

The 5 ns duration pulses were delivered by a nanosecond pulse generator, designed and fabricated by Transient Plasma Systems, Inc. (Torrance, CA), to chromaffin cells using a pair of cylindrical, gold-plated tungsten rod electrodes (127 μm diameter) with a gap of 100 μm between the electrode tips as detailed in previous publications [6, 13]. After achieving the whole-cell recording mode by rupturing the cell plasma membrane, the NEP-delivering electrodes were positioned to a predefined "working" position 40 μm from the bottom of the coverslip by a motorized MP-225 micromanipulator (Sutter Instruments, Novato, CA), with the patched cell situated midway between the electrode tips. A single or pair of 5 ns pulses was delivered to the cells at E-field amplitudes ranging from 5 to 10 MV/m. The E-field distribution at the location of the target cell was computed by the Finite-Difference Time-Domain (FDTD) method using the software package SEMCAD X (version 14.8.5, SPEAG, Zurich,

Switzerland) as previously described [5, 6, 13]. In all experiments, two sham (control) exposures preceded the delivery of a single or twin NEP(s) controlled by a program written in LabVIEW [5, 6, 13].

### Reagents

Ham's F-12 medium, dispase II and the antibiotics-antimycotics were obtained from Gibco Laboratories (Grand Island, NY, USA) and bovine calf serum was purchased from Gemini Bio-Products (West Sacramento, CA, USA). Collagenase B and methyl-β-cyclodextrin were purchased from MilliporeSigma (St. Louis, MO, USA). Tetrodotoxin was purchased from Alomone Labs (Jerusalem, Israel). All other chemicals were reagent grade and purchased from standard commercial sources.

### Data analysis

For all experiments, the results were obtained using cells from different days in culture and different cell preparations and presented as the mean ± standard error (SE). The data were analyzed with SPSS or Origin 2020 (v. 9.7, OriginLab, Northampton, MA) software using either an unpaired two-tailed t test when the means of two groups were compared, or a One-Way or Two-Way ANOVA test for repeated measures followed by Tukey *post hoc* multiple range tests in multiple group comparisons. $P < 0.05$ was considered statistically significant.

## Results

### Voltage-gated Na$^+$ channels are responsible for the early inward current

An initial series of experiments performed in bovine chromaffin cells exposed to normal K$^+$-based external (BSS) and internal solutions was carried out to determine the ionic nature of the early inward current elicited by depolarizing voltage clamp steps from a holding of –70 mV. Three families of membrane currents elicited by the voltage clamp protocol shown in Fig 1A are presented in Fig 1B. The mean peak inward current measured for each was plotted as a function of step potential as shown in Fig 1C. For cells in BSS, the early inward current activated near –30 mV, peaked around +10 mV and reversed at ~ +50 mV (Fig 1B and 1C). Total replacement of external Na$^+$ with the non-permeant NMDG$^+$ abolished the inward current (Fig 1B and 1C), which confirms the results of a recent study by our group performed under similar conditions [6] and demonstrates that Na$^+$ was the charge carrier responsible for this voltage-dependent inward current. Finally, the specific voltage-gated Na$^+$ channel inhibitor tetrodotoxin (TTX; 5 μM) also eliminated the inward current (Fig 1B and 1C). Taken together, these results support the idea that even when recorded in normal K$^+$-based salt solutions, the early inward current was predominantly carried by Na$^+$ and reflects the activity of TTX-sensitive voltage-dependent Na$^+$ channels and will thus be referred to as I$_{Na}$.

### Comparison of the effects of single and twin NEPs on I$_{Na}$

A constant voltage clamp step protocol consisting of repetitive voltage steps to +10 mV (near the peak of the I-V relationship; see Fig 1C) applied every 3 s from a holding potential of –70 mV was performed to monitor the time-dependent changes in I$_{Na}$ before and after cell exposure to 5 ns pulses. I$_{Na}$ was recorded for 1 min before NEP exposure to establish a baseline and then for 9 min afterwards to monitor the response.

The results presented in Fig 2A show that in cells unexposed to a NEP, I$_{Na}$ ran down spontaneously ~ 6% over the course of 9 min, a phenomenon most likely attributable to a negative shift of the voltage-dependence of voltage-gated Na$^+$ channels when recorded using the whole-

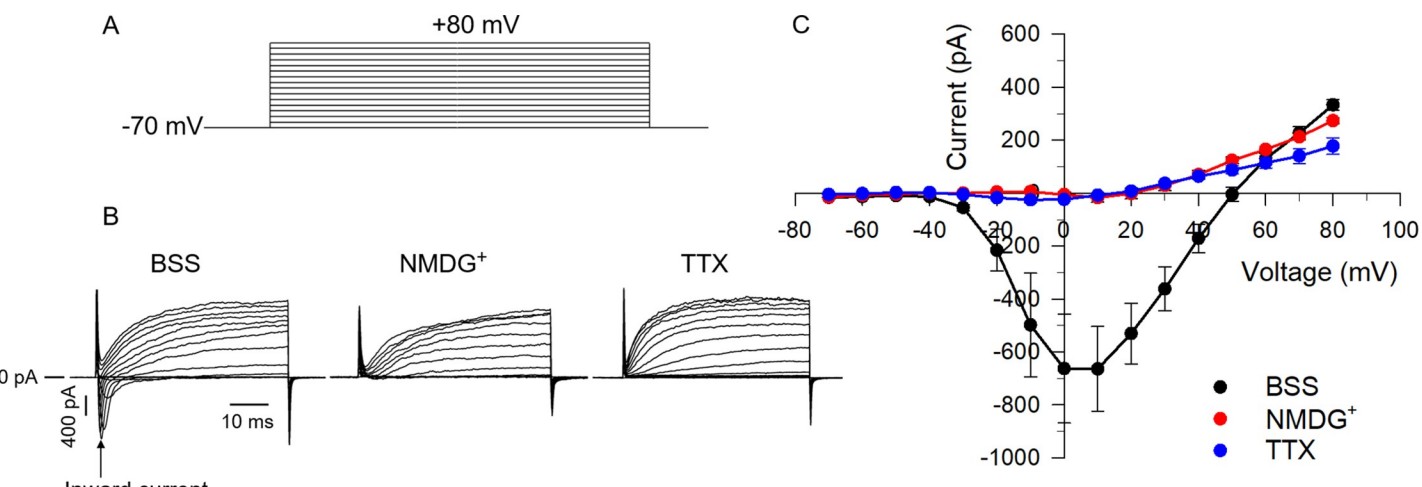

**Fig 1. Voltage-dependent properties of the early inward current recorded in whole-cell voltage clamped bovine chromaffin cells.** (A) The voltage clamp step protocol used to determine the voltage-dependence of peak inward current consisted of 50 ms steps ranging from –70 to +80 mV, applied in 10 mV increments every 2 s from a holding potential of –70 mV. (B) Representative families of membrane current traces elicited by the voltage step protocol shown in (A) recorded in different experimental conditions. BSS: normal K+-based external bath solution; NMDG+: an equimolar concentration of NMDG+ was used to replace Na+ in the external bath solution; TTX: 5 μM TTX was added to the BSS to block voltage-dependent Na+ channels. (C) I-V relationship for the peak inward current (arrow in B). Each data point represents the mean value ± SE (BSS, n = 4; NMDG+, n = 6; TTX, n = 3).

cell variant of the patch clamp technique [16]. A single 5 ns pulse applied at an E-field of 5 MV/m resulted in a rapid decrease of ~ 4% in peak $I_{Na}$ that then declined to a level that was ~ 89% of the initial level after 9 min (Fig 2A). These results are in agreement with those previously reported [6]. The application of a second 5 ns pulse 30 s after the first pulse amplified the

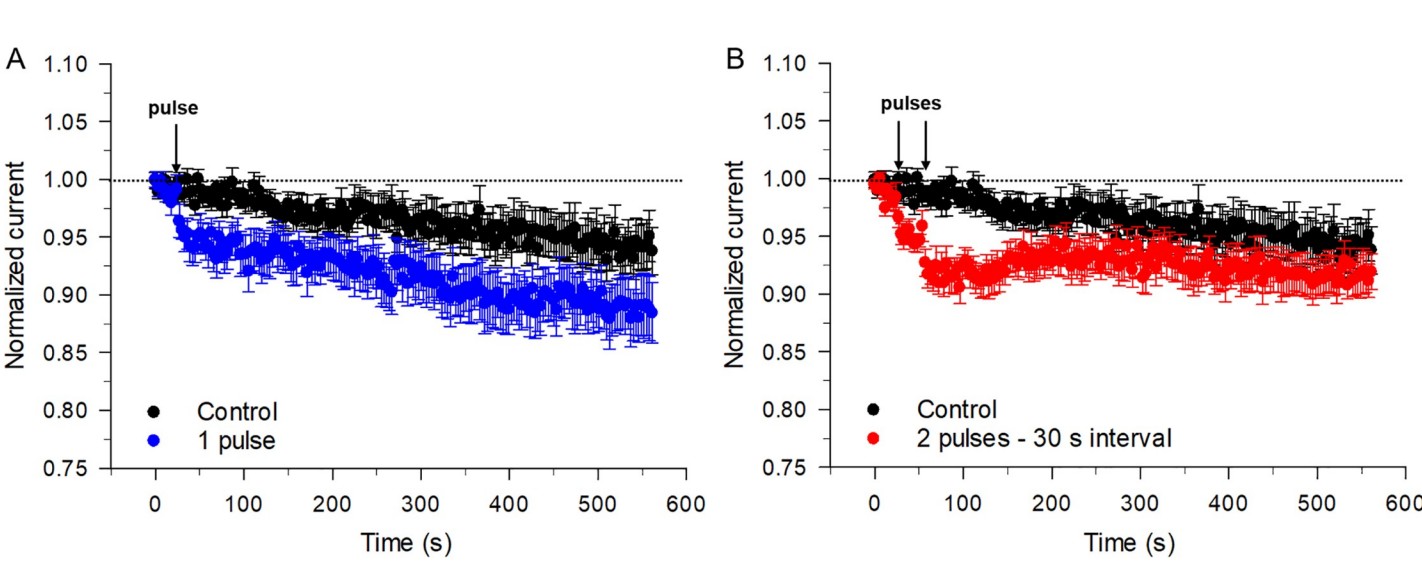

**Fig 2. Effects on $I_{Na}$ of a single or twin NEP applied at an E-field of 5 MV/m.** (A) Time course of changes in $I_{Na}$ for unexposed cells (control, black circles) compared to cells exposed to a single NEP (1 pulse, blue circles). (B) Time course of changes of $I_{Na}$ in cells exposed to a twin pulse applied with a 30 s interpulse interval (red circles) compared to that recorded in unexposed cells (control, black circles). For both panels, normalized current represents the magnitude of peak inward current evoked by the constant voltage step protocol described in the Methods normalized to the mean peak inward current recorded during the 8 voltage clamp steps to +10 mV (holding potential = –70 mV) that preceded the NEP(s). Each data point is a mean ± SE (control, n = 20; 1 pulse, n = 12; twin pulse, n = 10).

inhibitory effect of a single pulse on $I_{Na}$ from ~ 4% to 7% (Fig 2B). Representative traces of $I_{Na}$ are shown in S2 Fig. Interestingly and in contrast to the effects of a single NEP, the second pulse in the pair led to a progressive recovery of the current that stabilized near the level of $I_{Na}$ observed in control cells unexposed to NEPs (Fig 2B).

Decreasing the time interval between the twin pulses from 30 to 5 s (Fig 3A) again produced a similar inhibitory effect (~ 7%) on $I_{Na}$. However, the recovery of $I_{Na}$ was accelerated and enhanced relative to intervals between pulses that are longer. Further shortening the time interval between the two pulses to 1 s or 100 ms progressively attenuated the inhibitory effect (~ 6% and ~ 5%, respectively) on $I_{Na}$ (Fig 3B and 3C). Additionally, when the time interval between the two pulses was decreased to 10 ms (Fig 3D), the inhibition of $I_{Na}$ was slightly less (~ 3%) than the effect of a single pulse on peak $I_{Na}$. Again, shorter intervals between pulses accelerated and amplified the recovery of $I_{Na}$. In fact, when the interval was only 10 ms, the current leveled off near that recorded at the beginning of the experiment, that is prior to the application of NEPs (Fig 3D). Representative traces of $I_{Na}$ are shown in S2 Fig. One-Way ANOVA test revealed a significant difference between the cells exposed to the twin pulses and single pulse (P< 0.001).

At the higher E-field amplitude of 8 MV/m, $I_{Na}$ rapidly decreased by ~ 9% following the application of a single pulse (Fig 4A) as reported previously [6]. Exposure to a second 5 ns pulse 30 s after the first pulse increased the inhibitory effect produced by a single NEP from ~ 9% to 14% (Fig 4B). Representative traces of $I_{Na}$ are shown in S2 Fig. Decreasing the time interval between the twin pulses from 30 s to 5 s or 1 s led to a comparable inhibitory effect on $I_{Na}$ (~ 14%). For both intervals, $I_{Na}$ recovered to a level that was similar to that achieved by a single NEP. Decreasing the interval between the two pulses from 1 s to 100 ms (Fig 4C) led to a reduction of the inhibition from ~ 14% to 11%. Furthermore, when the time interval between the two pulses was decreased to 10 ms (Fig 4D and S2 Fig), the inhibition of $I_{Na}$ was slightly less (~ 10%) than that caused by the 100 ms time interval, which was still larger than the effect of a single pulse on peak $I_{Na}$. Similar to the effects of twin NEPs at 5 MV/m, intervals shorter than 1 s between pulses accelerated and increased the magnitude of the recovery of $I_{Na}$, which stabilized above that produced by a single NEP at 8 MV/m (Fig 4C and 4D). One-Way ANOVA test showed a significant difference between the cells exposed to the twin pulses and single pulse (P< 0.001).

A single NEP at 10 MV/m produced a 9% decrease in $I_{Na}$, which is comparable to that seen with a single pulse at an E-field of 8 MV/m (Fig 5A) as previously reported [6]. Once inhibited the current remained stable for ~ 9 min post-NEP. These results indicate that the effect of a single NEP on $I_{Na}$ saturated at the higher E-field amplitudes of 8 and 10 MV/m [6].

The time course of changes in $I_{Na}$ resulting from twin NEP exposures are shown in Fig 5. Increasing the E-field amplitude to 10 MV/m increased the inhibitory effect of twin NEPs on peak $I_{Na}$ (~ 15%) regardless of the interval between twin NEPs. At this E-field, the second pulse in the pair applied 10 ms to 30 s after the first pulse consistently increased the inhibitory effect of a single NEP from ~ 9% to ~ 15%. Representative traces of $I_{Na}$ are shown in S2 Fig. This contrasts with the effects of twin pulses at lower E-fields ($\leq$ 8 MV/m), which produced less inhibitory effects when the interval was progressively shortened (Figs 3 and 4). The data in Fig 5 also reveal that although the inhibitory effect of twin NEPs on $I_{Na}$ was still apparent at this E-field, current recovery was severely impaired, never reaching the level produced by a single NEP. Taken together, these results indicate that the effects of twin NEPs on $I_{Na}$ are highly dependent on the interval between the pulses and E-field magnitude. One-Way ANOVA test revealed a significant difference between the cells exposed to the pulse pair and single pulse (P< 0.001).

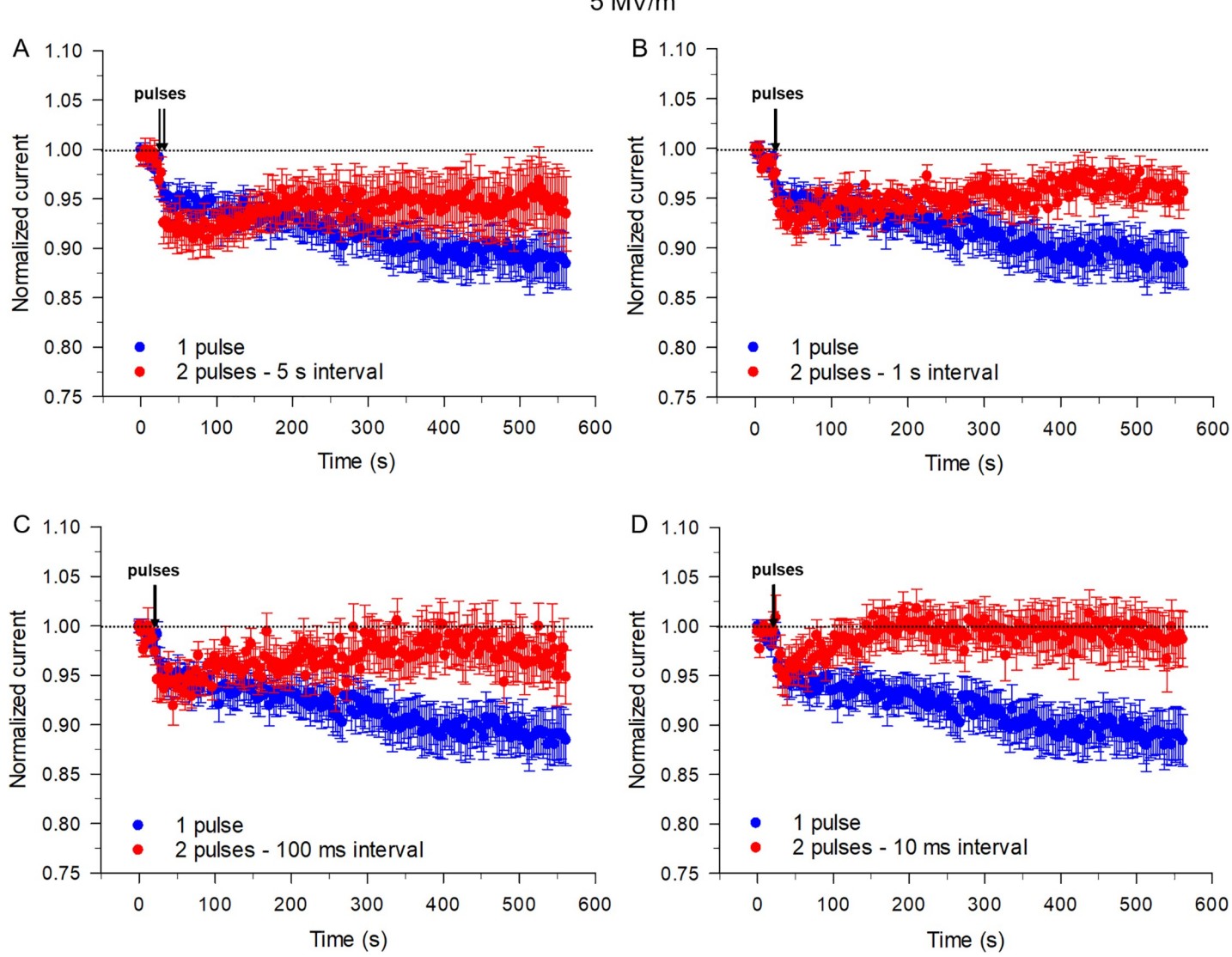

**Fig 3. Effects on $I_{Na}$ of the time interval between twin NEPs applied at an E-field of 5 MV/m.** Time course of changes in $I_{Na}$ for cells exposed to a single pulse (blue circles, data reproduced from Fig 2A) compared to cells exposed to twin NEPs (red circles) separated by a 5 s (A), 1 s (B), 100 ms (C) or 10 ms time interval (D), respectively. Each data point is a mean ± SE (1 pulse, n = 12; 2 pulses with a 5 s interval, n = 9; 2 pulses with a 1 s interval, n = 9; 2 pulses with a 100 ms interval, n = 9; 2 pulses with a 10 ms interval, n = 9).

Fig 6 summarizes the effects of E-field strength and interpulse interval on the inhibition and recovery of $I_{Na}$ in response to twin NEPs. As shown in Fig 6A and 6B, twin pulses applied at the E-fields of 5 and 8 MV/m, respectively, produced an additive inhibitory effect on $I_{Na}$ compared to that elicited by a single pulse when the interval between the pulses was longer than 100 ms. The additional inhibition was progressively reduced when the interval between pulses was 100 ms and 10 ms. At the E-field of 10 MV/m (Fig 6C), a twin pulse caused a consistently larger inhibitory effect relative to a single pulse, the magnitude of which was independent of pulse interval in the range of 10 ms to 30 s.

With respect to the level of recovery of $I_{Na}$ determined 9 min after NEP exposure, a single pulse applied at an E-field magnitude of 5 MV/m resulted in a decline in $I_{Na}$ to a level that was ~ 89% of the initial level compared with control cells in which $I_{Na}$ was ~ 94% of the initial level

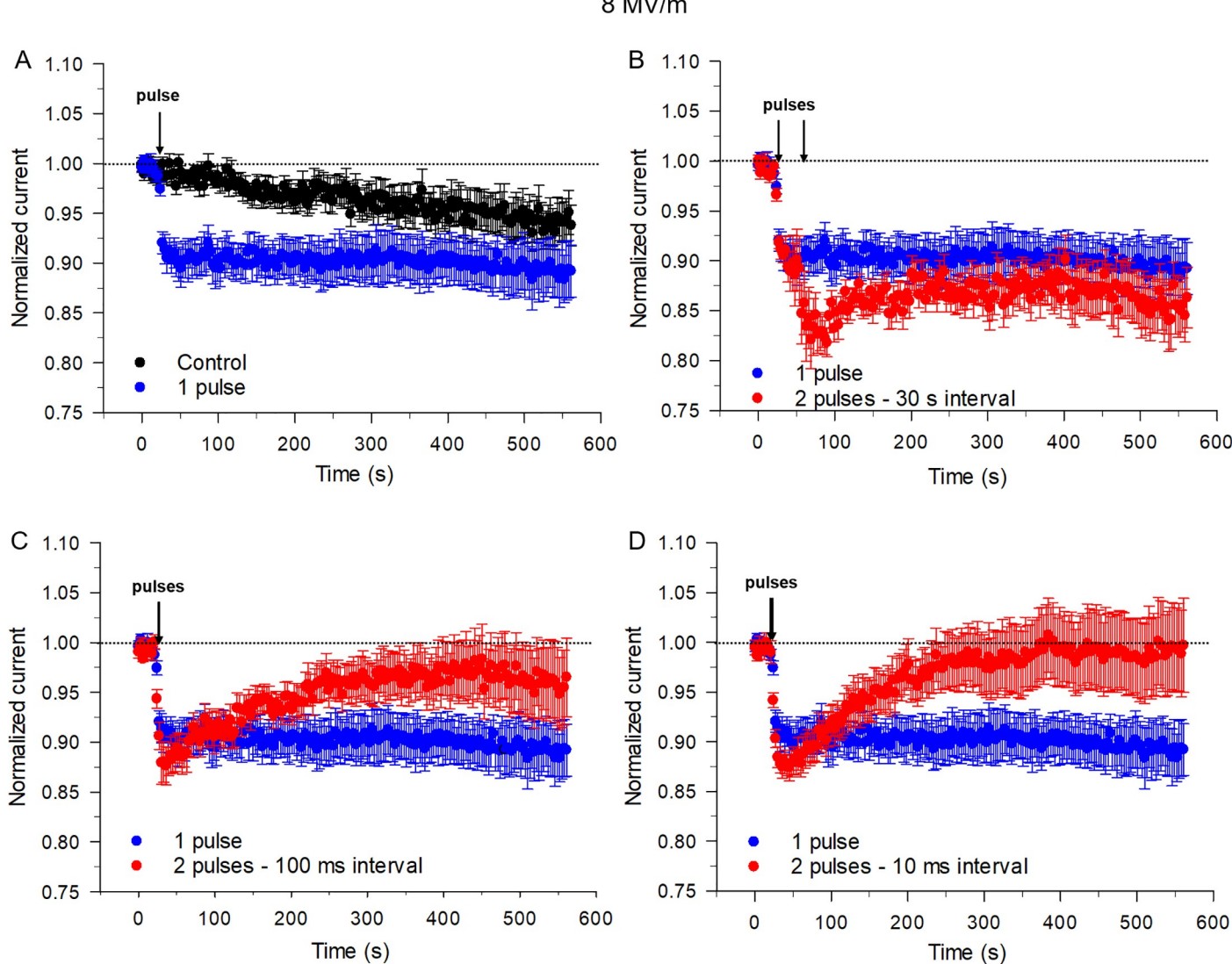

**Fig 4. Effects on $I_{Na}$ of the time interval between twin NEPs applied at an E-field of 8 MV/m.** (A) Time course of changes in $I_{Na}$ for unexposed cells (control, black circles) compared to cells exposed to a single pulse (blue circles). Time course of changes of $I_{Na}$ in cells exposed to a single pulse (blue circles) compared to cells exposed to twin pulses (red circles) with a 30 s (B), 100 ms (C) or 10 ms time interval (D), respectively. Each data point is a mean ± SE (control, n = 20; 1 pulse, n = 15; 2 pulses with a 30 s interval, n = 5; 2 pulses with a 100 ms interval, n = 10; 2 pulses with a 10 ms interval, n = 10). The control data set in panel A (black circles) are the same as those shown in Fig 2.

(Fig 6D). Paradoxically, the application of twin pulses at 5 MV/m (Fig 6D) or 8 MV/m (Fig 6E) led to a recovery of $I_{Na}$ after the inhibition that was inversely proportional to the time interval between the twin NEPs. In fact, at intervals of 10 or 100 ms, $I_{Na}$ magnitude was similar to or slightly greater than the level reached in cells unexposed to NEPs. Interestingly, the application of a twin pulse at an E-field of 10 MV/m (Fig 6F) did not lead to recovery of $I_{Na}$, which stabilized below the level reached with a single pulse regardless of the interval between the twin pulses.

## The application of twin NEPs led to similar effects on $I_{Na}$ when blocking $K^+$ and $Ca^{2+}$ currents

The experiments described above examining the effects of NEPs on $I_{Na}$ recorded at +10 mV were performed under physiological conditions with normal transmembrane gradients of

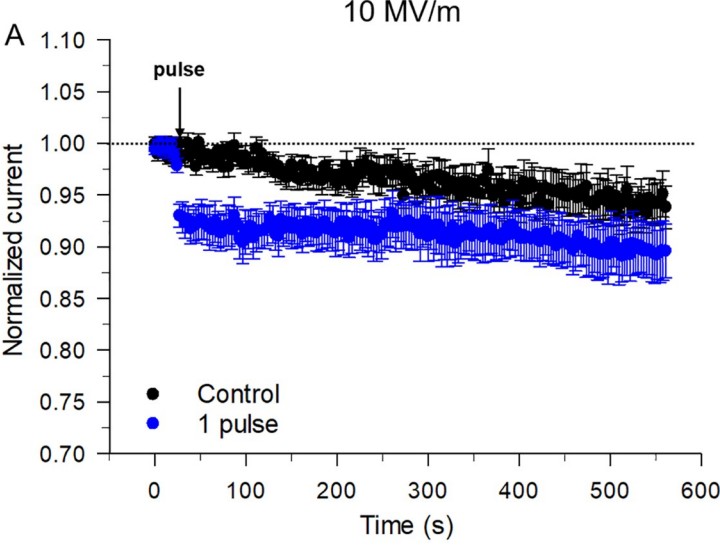

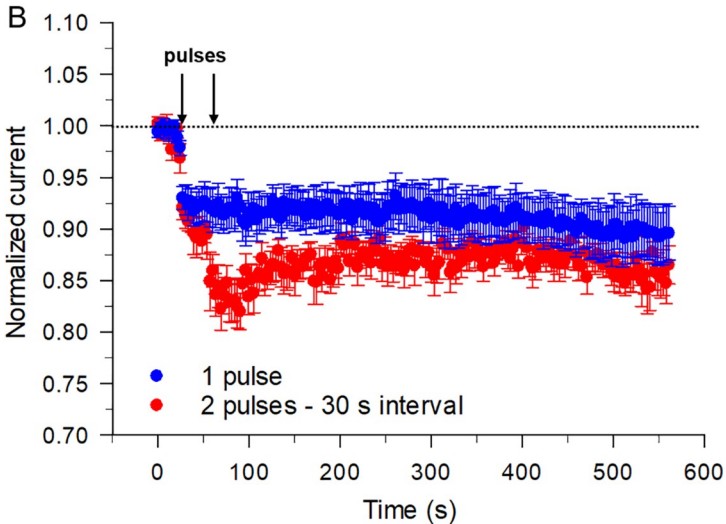

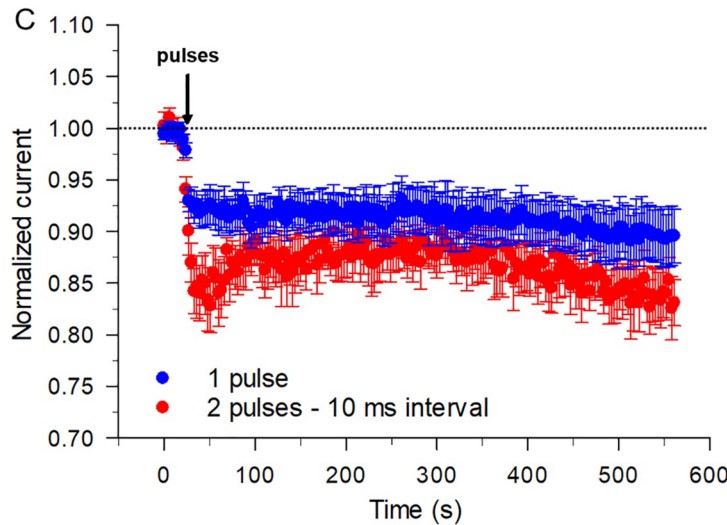

**Fig 5. Effect of a single pulse and twin NEPs applied at different time intervals on $I_{Na}$ at an E-field of 10 MV/m.**
(A) Time course of changes in $I_{Na}$ for unexposed cells (control, black circles) compared to cells exposed to a single pulse (blue circles). Time course of changes in $I_{Na}$ for cells exposed to a single pulse (blue circles) compared to cells exposed to twin pulses (red circles) with a 30 s (B) or 10 ms time interval (C), respectively. Each data point is a mean ± SE (control, n = 20; 1 pulse, n = 15; 2 pulses with a 30 s interval, n = 8; 2 pulses with a 10 ms interval, n = 7). The control data set in panel A (black circles) are the same as those shown in Fig 2.

$Na^+$, $K^+$ and $Ca^{2+}$. To isolate the effect of the NEPs on $Na^+$ channels, additional experiments were carried out by suppressing $K^+$ and $Ca^{2+}$ currents. First, $K^+$ currents were inhibited by replacing $K^+$ with $Cs^+$ and TEA both in the internal and external solutions, and by replacing K-gluconate with Cs-aspartate in the internal solution. An equimolar concentration of $Mg^{2+}$ was used to replace $Ca^{2+}$ in the external bath solution to suppress $Ca^{2+}$ currents instead of using the dihydropyridine antagonist nitrendipine because this compound also reduced $I_{Na}$ as previously reported by others [17] at the high concentration (20 μM) that is necessary to fully inhibit $Ca^{2+}$ currents in chromaffin cells [18]. As shown in Fig 7A, the activation threshold potential and magnitude of the inward current recorded between –30 to +40 mV were not affected by using $K^+$- and $Ca^{2+}$-free solutions; however, a small positive shift of the reversal potential (~ +5 mV) was noted, which can be attributed to a small but significant contribution from outward $K^+$ currents due to the reduced driving force for $Na^+$ at higher voltages (> +40 mV). Application of twin pulses with a 10 ms interpulse interval applied at an E-field of 8 MV/m produced a similar inhibitory effect on $I_{Na}$ shortly after pulse application, which recovered following a nearly superimposable time course to that recorded in regular BSS (unpaired t test; P = 0.7; Fig 7B), confirming the results obtained with normal $K^+$-based external and internal solutions (e.g. Fig 4D) and that the paradoxical effects of twin pulses were targeted to $I_{Na}$.

## Twin NEPs reduced the maximal conductance but not the voltage-dependence of $I_{Na}$

The inhibitory and reversal effects of twin 5 ns pulses on $I_{Na}$ could be caused by a shift of the steady-state inactivation or activation curve, or both. In order to test for possible changes in steady-state inactivation, a typical double-pulse protocol was used whereby the cells were held over a range of potentials for 1 s and then depolarized to a test pulse (+10 mV) to evaluate the availability of $Na^+$ channels. As for Fig 7, all these experiments were carried out under conditions isolating $I_{Na}$. Fig 8A shows representative traces of $I_{Na}$ recorded just prior (left), immediately after (middle) and 9 min following (right) the application of twin NEPs. Fig 8B shows the effects of twin pulses with a 1 s interpulse interval at E-field strength of 8 MV/m on the voltage-dependence of inactivation. This interval and E-field were chosen because we started to observe the largest inhibition of $I_{Na}$ with these pulse parameters (Fig 6E). Half-maximal inactivation voltages $V_{0.5}$ were –50.4 ± 1.2 mV ($k$ = 8.3 ± 0.6), –51.3 ± 0.7 mV ($k$ = 7.1 ± 0.4) and –54.4 ± 0.6 mV ($k$ = 8.1 ± 0.4) before the pulses, immediately after and 9 min after the pulses, respectively. These data show that there was no significant effect of the twin pulses on the steady-state inactivation of $I_{Na}$ (One-Way ANOVA, P > 0.05).

The three families of $I_{Na}$ shown in Fig 8A, which were recorded in the same cell just prior, immediately after and 9 min following the application of twin NEPs with an interval of 1 s at 8 MV/m, show the inhibitory effect of twin pulses on $I_{Na}$ and partial recovery of the current over a 9 min post-pulse period. Fig 8C shows the I-V relationships of $I_{Na}$ recorded for these same time periods. As previously reported by our group for a single NEP [6], $I_{Na}$ was significantly reduced by twin NEPs over a wide range of membrane potentials. Twin pulses also produced a significant negative shift (~ –6 mV) of the reversal potential relative to the control I-V obtained prior to pulse delivery. We hypothesized that this shift might be attributable to a relatively

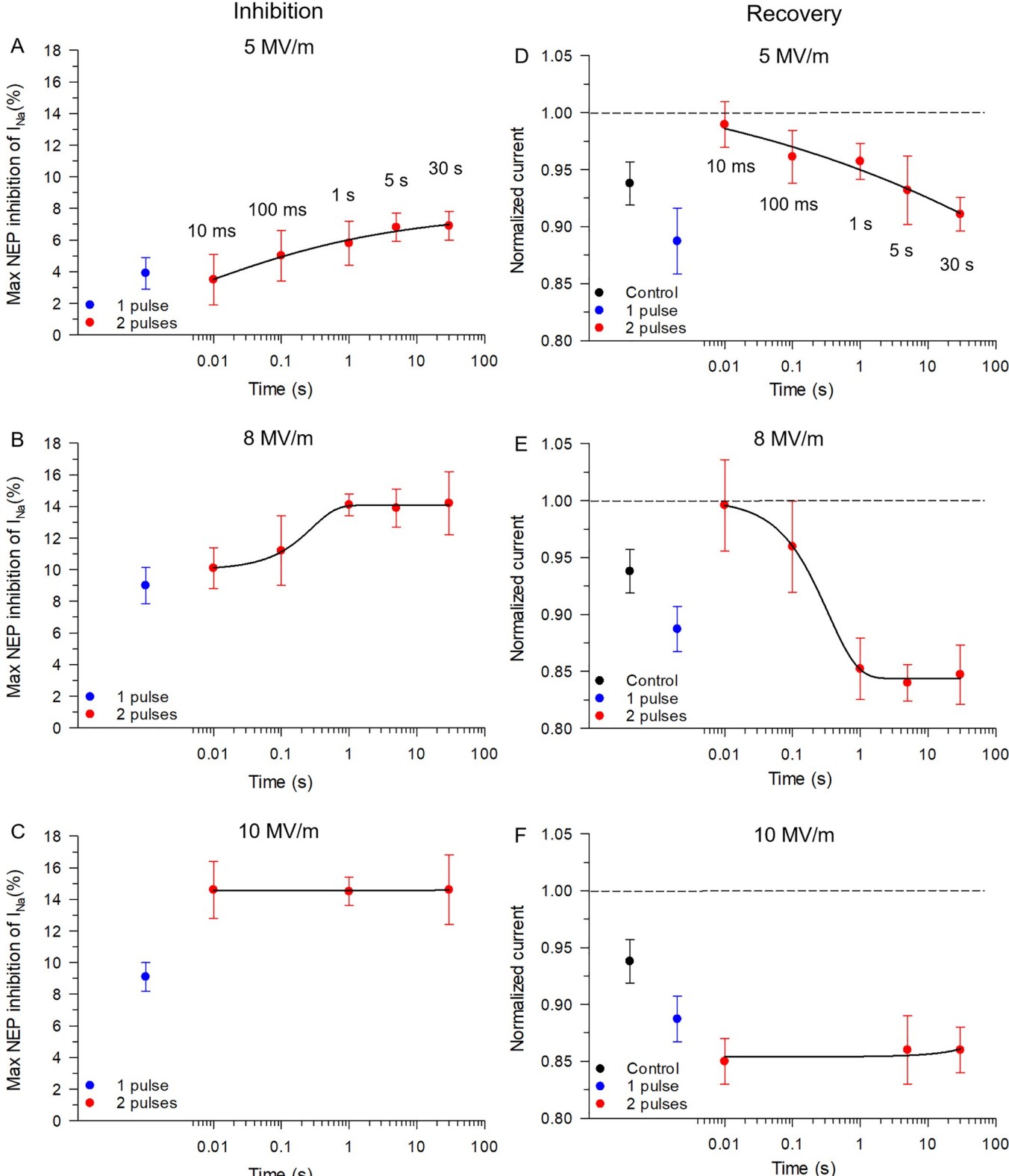

**Fig 6. Time interval and E-field strength dependence of the inhibitory and recovery effect of twin NEPs on $I_{Na}$.** For the left panel (inhibitory), each semi-logarithmic plot reports the mean ± SE % inhibition of peak inward Na$^+$ current elicited in cells exposed to a single pulse (blue circles), or cells exposed to twin NEPs (red circles) as a function of the interval between the pulse pair at an E-field of 5 MV/m (A), 8 MV/m (B) or 10 MV/m (C), respectively. For the right

panel (recovery), each plot reports the mean ± SE normalized peak inward Na$^+$ current registered in unexposed cells (control, black circles), cells exposed to a single pulse (blue circles), or cells exposed to twin NEPs (red circles) as a function of the interval between the pulse pair at an E-field of 5 MV/m (D), 8 MV/m (E) or 10 MV/m (F), respectively. In panels D, E and F, all measurements were performed 9 min after the application of the twin pulses. In panels A and D, the solid line is a logistic fit to the data points. The solid lines in panels B and E are a sigmoidal fit to the data points. In panels C and F, the solid line is a linear fit. Results were obtained from the same cells as those shown in Figs 2, 3, 4 and 5.

small leak current (I$_{leak}$; Fig 8A, middle family of traces) elicited by twin NEPs an observation consistent with the concept of membrane electronanoporation [5, 6]. To determine the impact of this basal conductance on peak I$_{Na}$ measurements, I$_{leak}$ was estimated in separate experiments using K$^+$-free pipette and bath solutions containing Cs$^+$ and TEA, which were used to suppress voltage-gated K$^+$ channels, and 20 μM nitrendipine applied to the bath to inhibit voltage-gated Ca$^{2+}$ channels [18]. A 2 s voltage ramp protocol spanning from –70 to +80 mV was used to generate I-V relationships before and after the application of twin pulses with a 1 s interval at E-field of 8 MV/m. Voltage-gated Na$^+$ channels were voltage-inactivated by this slow voltage ramp protocol. Fig 8C depicts the small NEP-sensitive I-V for the leak current (I$_{leak}$; grey circles; obtained by digital subtraction of the control current recorded prior to the application of twin NEPs), which displayed mild inward rectification and a reversal potential near 0 mV. Leak current measurements were performed by averaging the current during the voltage ramp for a range spanning ± 0.5 mV at each target potential (used for peak I$_{Na}$ measurements). Subtraction of the I-V of I$_{leak}$ from the I-V of I$_{Na}$ recorded immediately after delivery of twin NEPs yielded a "corrected" I-V for I$_{Na}$ (green circles) that displayed a very similar amplitude over the entire range of membrane potentials and a reversal potential (~ +62 mV) that coincided with the control I-V for I$_{Na}$ recorded prior the application of twin NEPs. This correction procedure was not necessary for I$_{Na}$ measured 9 min after the NEPs were applied because I$_{leak}$ slowly disappeared over this time span, which is similar to our previous report with a single NEP [5]. Taken together, our data demonstrate that peak I$_{Na}$ measured at +10 mV, which is the test potential mostly used in our study, was nearly superimposable to the uncorrected I$_{Na}$, indicating that the changes in peak Na$^+$ current mediated by NEPs were minimally contaminated by the presence of I$_{leak}$.

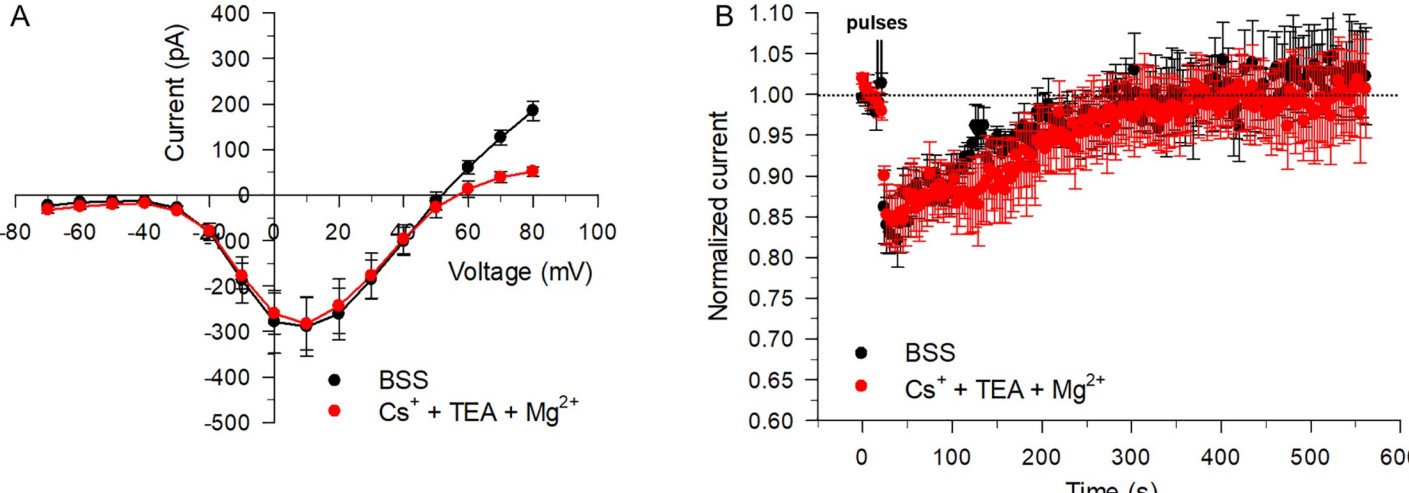

**Fig 7. Effect of twin pulses on I$_{Na}$ recorded in K$^+$- and Ca$^{2+}$-free BSS at an E-field of 8 MV/m.** (A) Effect of eliminating K$^+$ and Ca$^{2+}$ currents on the I-V relationship of the early peak inward current. I-V relationships were obtained either in regular BSS (black circles) or in K$^+$- and Ca$^{2+}$-free BSS (Cs$^+$ + TEA + Mg$^{2+}$; red circles). (B) Mean time course of changes of peak I$_{Na}$ recorded with normal (BSS) or Cs$^+$-TEA-Mg$^{2+}$-based solutions in cells exposed to twin pulses (8 MV/m) with a 10 ms interpulse interval. Data are expressed as the mean ± SE (BSS, n = 5; Cs$^+$ + TEA + Mg$^{2+}$, n = 6).

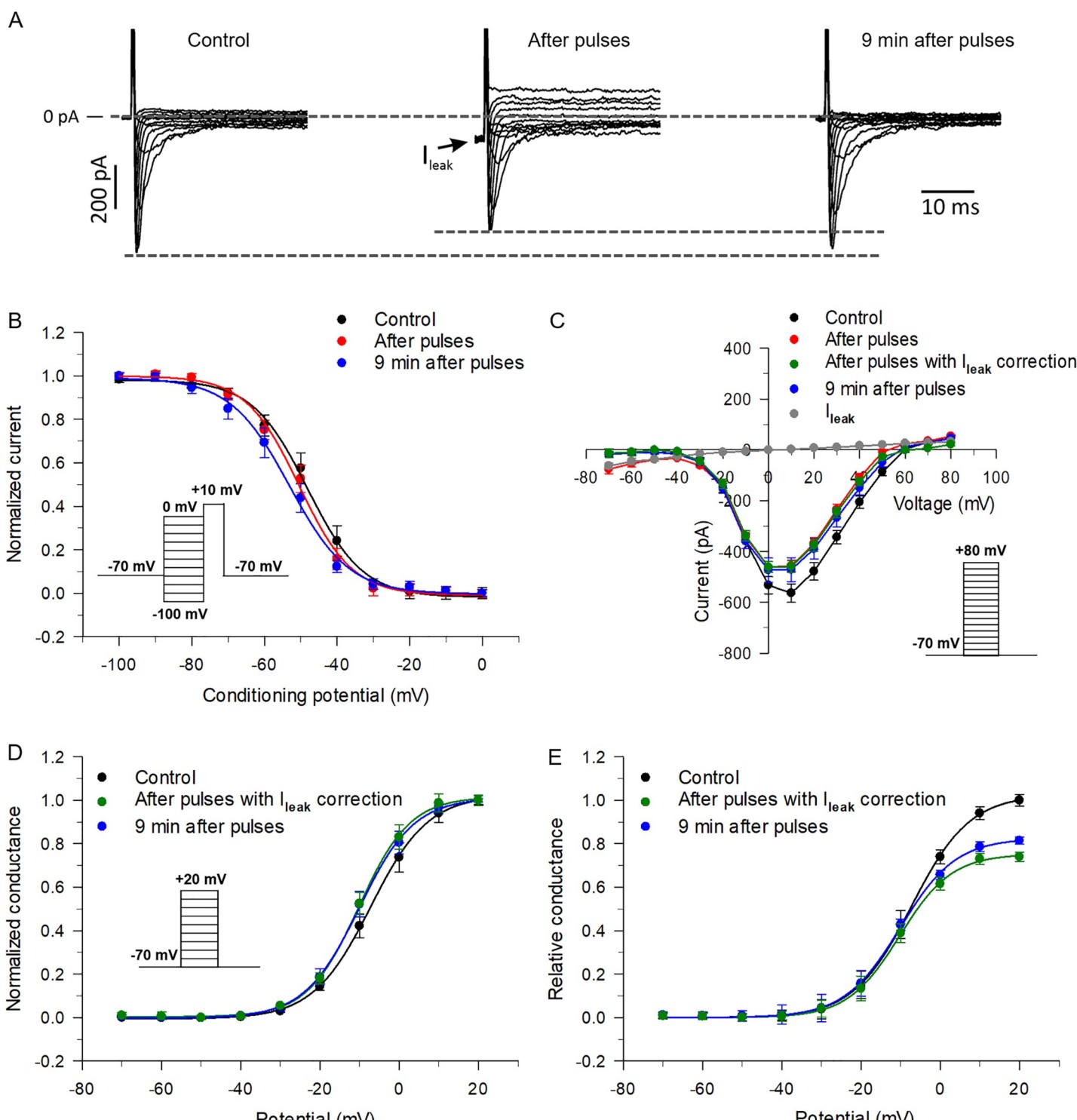

**Fig 8. Effects of twin NEPs applied with a 1 s time interval on the voltage-dependence of steady-state inactivation, activation and conductance of $I_{Na}$ at an E-field of 8 MV/m.** (A) Three families of $I_{Na}$ recorded in the same cell from a typical experiment; as depicted, these families of traces were recorded before (Control; left traces), immediately after (After pulses; middle traces) and 9 min after (right traces) twin pulse exposure. For each family of traces, step potentials spanned from −40 to +60 mV. (B) Steady-state inactivation curves of $I_{Na}$ recorded before (control, black circles), right after (red circles) and 9 min after (blue circles) twin pulse exposure. $I_{Na}$ in each case was normalized to its respective maximal value, plotted as a function of the conditioning potential and fitted to a Boltzmann function (solid lines). The data represent the mean value ± SE, n = 5. The $V_{0.5}$ and slope factor values derived from all Boltzmann fits can be found in the text. Overall One-Way ANOVA yielded P values for $V_{0.5}$ (P = 0.24373) and $k$ (P = 0.8831) that revealed no significant differences between the means of the three groups. (C) I-V relationships recorded before

(control, black circles), immediately after (red circles), and 9 min after (blue circles) twin pulse exposure; the I-V relationship labeled with green circles ("After pulses with correction") was obtained by subtraction of the I-V of the NEP-induced leak current (grey circles; $I_{leak}$) from that measured after delivery of the NEPs (red circles). $I_{leak}$ was measured in separate experiments using a voltage ramp protocol and $K^+$-free external solution containing 20 μM nitrendipine as described in the Methods. All data points are expressed as the mean ± SE (I-V, n = 5; $I_{leak}$, n = 4). (D) Steady-state activation curves of $I_{Na}$ recorded before (control, black circles), immediately after with correction (green circles) and 9 min following (blue circles) twin pulse exposure. For each data set, $I_{Na}$ in each case was normalized to its respective maximal conductance (calculated as indicated in the Methods), plotted as a function of the test potential and fitted to a Boltzmann function (solid lines). All data points are expressed as the mean ± SE (n = 5). The $V_{0.5}$ and slope factor values derived from all Boltzmann fits can be found in the text. Overall One-Way ANOVA yielded P values for $V_{0.5}$ (P = 0.18424) and $k$ (P = 0.46385) that revealed no significant differences between the means of the three groups. (E) Relative conductance, calculated by normalizing the conductance estimated before and after the pulses to the maximum conductance before the pulses, was plotted against test potential. Peak $I_{Na}$ was converted to conductance as described in the Methods. The data were fitted to a Boltzmann function (solid lines). All data points are expressed as the mean ± SE (n = 5). An overall Two-Way ANOVA test revealed significant differences between Pulse treatment (P < 0.001), Voltage (P < 0.001) and the Interaction between Pulse treatment and Voltage (P = 0.00679). Overall Tukey *post hoc* tests revealed a significant difference between "Control" and "After pulses with $I_{leak}$ correction" (P < 0.001) and between "Control" and "9 min after pulses" (P = 0.00616), but no significant difference between "After pulses with $I_{leak}$ correction" and "9 min after pulses" (P = 0.84134). Tukey *Post hoc* tests only showed significant differences between "Control" and "After pulses with $I_{leak}$ correction" at +10 (P = 0.04208) and +20 mV (P < 0.001), and between "Control" and "9 min after pulses" at +20 mV (P = 0.01159).

We next examined whether twin NEPs affect the steady-state activation curves of $I_{Na}$ using the data presented in Fig 8C. These curves were constructed by estimating the normalized $Na^+$ conductance ($G_{Na}/G_{max}$) from measurements of corrected peak inward $Na^+$ current evoked by the depolarizing steps used to build the I-V curve and the driving force ($V–E_{rev}$) as described in the Methods. As for the steady-state inactivation curve, our results indicate that twin pulses with a 1 s interval at an E-field of 8 MV/m produced no significant effect on the voltage-dependence of activation of $I_{Na}$ (Fig 8D). Half-maximal activation voltages $V_{0.5}$ were –8.2 ± 0.5 mV ($k$ = 6.3 ± 0.2), –10.7 ± 0.5 mV ($k$ = 6.6 ± 0.2) and –10.5 ± 0.5 mV ($k$ = 6.3 ± 0.3) before, immediately after with correction and 9 min after pulse delivery, respectively. One-Way ANOVA tests revealed no significant differences between the three groups for both parameters (P > 0.05). The $E_{rev}$ (derived from the I-Vs in Fig 8C) were 59.6 ± 2.6 mV, 62.1 ± 2 mV (corrected) and 57.6 ± 1.6 mV before, immediately after and 9 min after the pulses, respectively, which were also not significantly different from each other (P > 0.05).

Another possibility that could account for the effects of twin NEPs on peak $I_{Na}$ is a change in maximal conductance of the $Na^+$ channel, as previously reported for a single pulse [6]. To assess this possibility, we determined the absolute $Na^+$ conductance before and after twin pulses with a 1 s interpulse interval at an E-field of 8 MV/m. For this determination, relative $Na^+$ channel conductance was calculated by normalizing the conductance of the channel after the pulses to the conductance of the channel measured before the pulses. As shown in Fig 8E, the maximal conductance of $I_{Na}$ estimated immediately after the twin pulses was significantly reduced at +10 and +20 mV (P < 0.05). For example, at +20 mV, the decrease in maximal $Na^+$ conductance was ~ 26% after the pulses. However, 9 min after the pulses the maximal $Na^+$ conductance did not significantly recover, an observation consistent with the lack of recovery of $I_{Na}$ after the twin pulse-induced inhibition with longer intervals as described above in Fig 6D and 6E.

## Disruption of lipid rafts alters the effects of twin NEPs on $I_{Na}$

It is well known that lipid rafts and one of its subset structures called caveolae are specialized membrane assemblies involved in compartmentalized signaling and protein trafficking [19]. In this regard, TTX-resistant $Na_V1.5$ and $Na_V1.8$ channels were respectively shown to be clustered in caveolae in breast cancer cells and sensory dorsal root ganglion neurons [20, 21]. Although it has not yet been established whether TTX-sensitive $Na_V1.7$ channels are similarly clustered in lipid rafts, we nevertheless hypothesized that twin NEPs may be altering the function of voltage-gated $Na^+$ channels in chromaffin cells by indirectly perturbing the composition and/or structure of lipid rafts regulating their activity and recycling. Methyl-β-

cyclodextrin (MβCD), an agent known to deplete membrane cholesterol and impair lipid rafts and caveolae [19, 22], was used to investigate whether the disruption of lipid rafts altered the response of $I_{Na}$ to twin NEPs. Cells were pre-incubated for a minimum of 30 min at a concentration of 3 mg/ml MβCD. We first tested the effects of MβCD on $I_{Na}$ in cells unexposed to NEPs. As shown in Fig 9A, MβCD produced a 42% inhibition of the inward current at the test potential of +10 mV and caused a small negative shift of the reversal potential (~ 5 mV). This small shift in reversal potential is probably attributable to a relatively greater contribution from outward voltage-dependent $K^+$ currents triggered by the MβCD-induced inhibition of peak $Na^+$ current measured at voltages where the driving force for $Na^+$ is diminished. These results imply that $Na^+$ channels in chromaffin cells may too be regulated by lipid rafts in a manner suggestive of a permissive role.

We next examined how depleting membrane cholesterol affected the inhibition of $I_{Na}$ in cells exposed to a single 5 ns pulse at the E-field of 8 MV/m that caused a ~ 9% drop in $Na^+$ conductance, followed by a stabilization of the current (Fig 4A). As shown in Fig 9B, pretreating cells with MβCD minimally altered the response to the pulse, reducing slightly the inhibitory effect of the pulse on $I_{Na}$, and causing a minor increase in the level of recovery of the current that was just at the limit of statistical significance (unpaired t test P = 0.07). However, MβCD pretreatment of the cells impacted the effect of twin NEPs on the inward current in a different manner. In cells pretreated with MβCD, a pair of NEPs at an E-field of 5 MV/m with a 10 ms interval between pulses increased the inhibitory effect on $I_{Na}$ from ~ 3% to ~ 5% (Fig 9C). Interestingly, the recovery of $I_{Na}$ in MβCD-treated cells was reduced but not significantly relative to that registered from cells unexposed to MβCD (unpaired t test P = 0.19). The effects noted at an E-field of 5 MV/m were similar but much more prominent at an E-field of 8 MV/m (Fig 9D). The inhibition caused by twin NEPs was enhanced from ~ 14% to ~ 22% in cells pretreated with MβCD compared to untreated cells. Moreover, the recovery of $I_{Na}$ was nearly abolished by cholesterol depletion (unpaired t test P < 0.001).

## Discussion

Stimulation of adrenal chromaffin cells with 5 ns pulses is effective for evoking neurosecretion in the absence of overt deleterious effects [3, 4, 23, 24]. As reported in this study, we found that application of 5 ns pulses as a pulse pair (twin pulse) can modulate voltage-gated $Na^+$ channels in a unique manner, which points to the potential for NEPs to influence overall cell excitability in novel ways.

### The early rapid inward current is mediated by voltage-gated $Na^+$ channels

In this study, we investigated in whole-cell voltage clamped bovine chromaffin cells the effects of twin nanosecond pulses on the activity of the early and rapid inward current recorded in physiological gradients of $Na^+$, $K^+$ and $Ca^{2+}$ across the membrane. Our data showed that this current was largely carried by $Na^+$ influx across voltage-gated $Na^+$ channels because: 1) the inward current was eliminated by external $Na^+$ substitution with the non-permeant $NMDG^+$, as similarly reported by our group [6], or 2) by the specific $Na_V$ inhibitor tetrodotoxin, 3) the voltage-dependence was similar to that of the main $Na^+$ channel pore-forming subunit expressed in chromaffin cells, $Na_V1.7$ [9–12], 4) the properties of the inward current were nearly identical to those of the inward current recorded in $K^+$-free solutions containing $Cs^+$ and TEA to inhibit $K^+$ channels, and nitrendipine applied externally to inhibit $Ca^{2+}$ channels, with the exception of the current measured near the reversal potential, and 5) the effects of twin NEPs on the inward current were very virtually superimposable whether the current was recorded using physiological solutions or solutions designed to isolate $I_{Na}$. A small exception

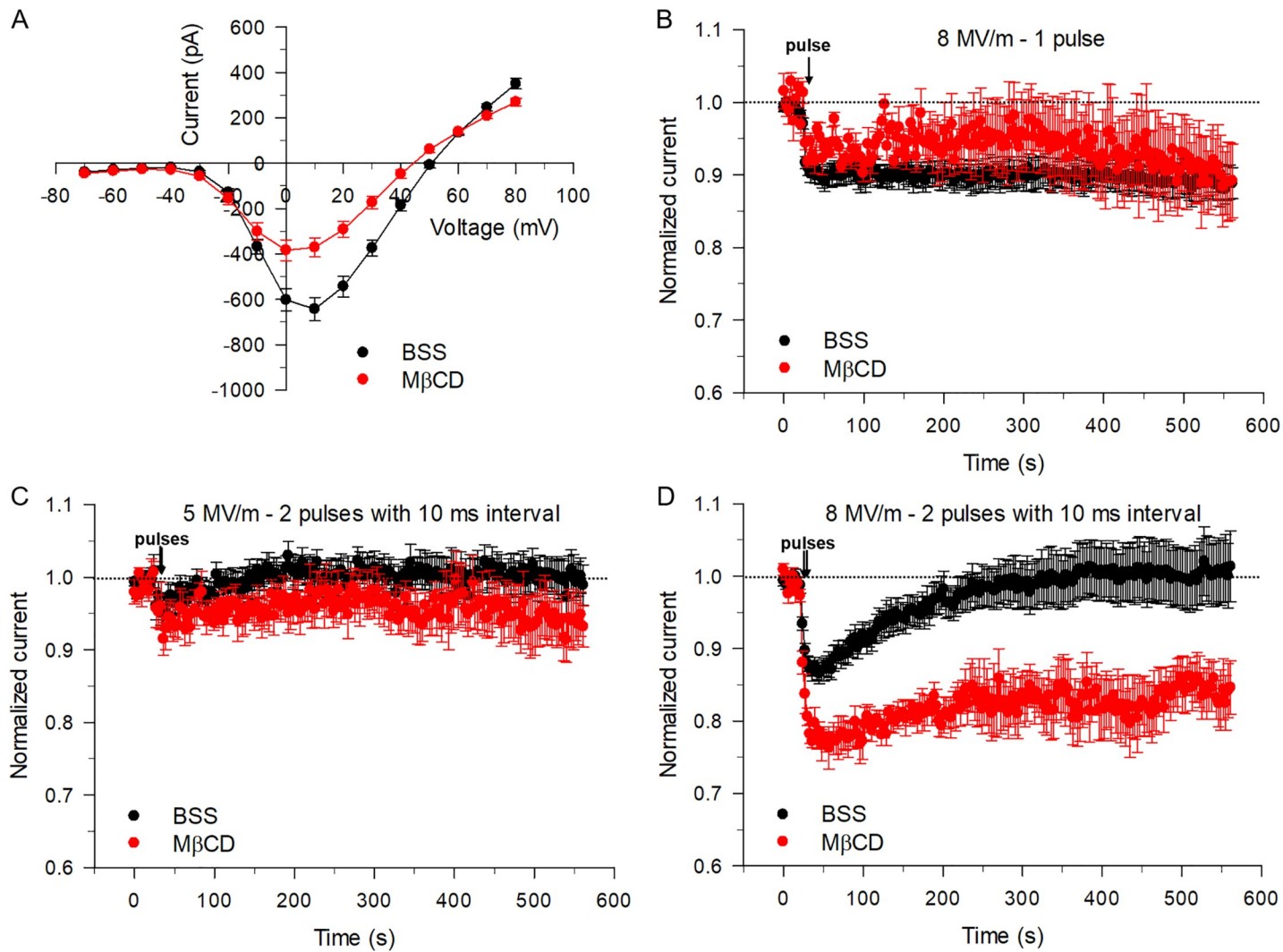

**Fig 9. Disruption of lipid rafts reduces $I_{Na}$ and alters the impact of twin NEPs on the inward current.** (A) The voltage step protocol to construct the I-V relationship of peak $I_{Na}$ was described in the Methods. The mean I-V relationships were obtained either in the presence of normal BSS (black circles) or in BSS containing MβCD (red circles). (B) Time course of changes in $I_{Na}$ for cells exposed to a single pulse at an E-field of 8 MV/m recorded in the absence (black circles) or presence (red circles) of MβCD. (C) Time course of changes in $I_{Na}$ for cells exposed to twin NEPs at an E-field of 5 MV/m with a 10 ms time interval between the pulses recorded in the absence (black circles) or presence (red circles) of MβCD. (D) Time course of changes in $I_{Na}$ for cells exposed to twin pulses at an E-field of 8 MV/m with a 10 ms time interval between the pulses recorded in the absence (black circles) or presence (red circles) MβCD. Cells exposed to MβCD were pre-incubated with the drug at a concentration of 3 mg/ml for a minimum of 30 min. Data are expressed as the mean ± SE. (A): BSS, n = 16; MβCD, n = 14; (B): BSS, n = 15; MβCD, n = 6; (C): BSS, n = 9; MβCD, n = 5; (D): BSS, n = 10; MβCD, n = 8).

was the contamination of the inward current recorded in normal solutions around the reversal potential, with some contamination from outward $K^+$ currents that shifted $E_{rev}$ by a few millivolts in the negative direction. This had no effect on our findings since the effects of twin NEPs were investigated at +10 mV. Moreover, a more detailed analysis of the effects of twin NEPs on $I_{Na}$ was performed under conditions that isolated $I_{Na}$. These arguments give credence to the notion that the fast inward current modulated by NEPs in this study was $I_{Na}$.

### Inhibitory effects of twin NEPs on $I_{Na}$

A previous study from our group reported that a single 5 ns pulse was sufficient to produce a small but consistent inhibition of $I_{Na}$ in bovine chromaffin cells [6]. This was reminiscent of

studies showing that much longer nanosecond pulses (300 ns) also reduced $I_{Na}$ in chromaffin cells [8]. Based on these findings, we tested in this study the hypothesis that the application of a second NEP of identical duration and E-field strength would produce an amplified inhibitory effect on $I_{Na}$. We showed that irrespective of the E-field tested (5 to 10 MV/m), the inhibition of $I_{Na}$ approximately doubled relative to that produced by a single pulse when the interval between the two pulses was long ($\geq$ 5 s). One puzzling observation was that for twin pulses at E-field strengths of 5 and 8 MV/m but not 10 MV/m, reducing the interval between the pulses decreased the additive inhibitory activity on $I_{Na}$. In fact, for 5 MV/m, the magnitude of the inhibition produced by twin pulses with a 10 ms interpulse gap was slightly less than that evoked by a single NEP. These data suggest that the process initiated by the first pulse was somehow interrupted or reversed by the second pulse in the pair. This is analogous to the phenomenon of bipolar cancellation whereby the application of a second nanosecond pulse of opposite polarity can attenuate or abolish a cellular response ($Ca^{2+}$ transient, electronanoporation, fluorescent dye uptake, etc.) triggered by the corresponding unipolar pulse [25–33].

As for a single NEP [6], the inhibitory effects of twin NEPs on $I_{Na}$ were not accompanied by a change in the voltage-dependence of steady-state activation and inactivation, indicating that single or twin NEPs did not appear to interfere with the voltage sensor. After correction for the presence of a small inwardly-rectifying basal conductance induced by a pulse pair, which was previously reported by our group in chromaffin cells for a single NEP [5], the reversal potential of the inward current was also unchanged by twin NEPs, suggesting that the ion selectivity and perhaps the ion conduction pathway remained intact. The only parameter that was altered by twin NEPs was a reduction in the maximal $Na^+$ conductance $G_{Max}$ and this observation was similar to that evoked by a single NEP in this and a previous study from our group [6].

For a macroscopic ionic current, $G_{Max}$ is defined by the following equation:

$$G_{Max} = g_{Na} \; x \; P_{oMax} \; x \; N$$

where $g_{Na}$ is the single-channel conductance (in pS), $P_{oMax}$ is the maximal open probability and $N$ the number of active channels in the membrane. It is likely that based on the lack of effect of twin NEPs on the reversal potential that the single-channel conductance, an indicator of ion flux across a single open ion channel, was minimally altered by twin pulses. On the basis of unperturbed voltage sensing by the channels, it is possible that $P_{oMax}$ may also have been unaffected by twin NEPs although this will require further testing.

One plausible hypothesis is that twin NEPs produced a transient suppression of the number of active channels through perturbations of the lipid bilayer closely interacting with the channel protein. This suggestion is based on a number of indirect observations from our group and others: 1) theoretical molecular dynamics studies [34–36] demonstrated that high intensity NEPs are capable of triggering the formation of hydrophilic nanopores in the phospholipid bilayer that lower the energy barrier for ion conduction across the membrane; 2) single or multiple NEPs of various durations are known to cause important perturbations of the lipid bilayer, a process called electronanoporation, that allows the membrane to become permeable not only to ions [37, 38] but in some cell types also to fluorescent organic molecules such as propidium iodide and YO-PRO-1 [25, 26, 29, 33, 38–42]. Even though the application of up to ten 5 ns pulses does not lead to YO-PRO-1 uptake in chromaffin cells [23], a non-selective membrane conductance ($I_{leak}$) permeable to $Na^+$ could be elicited by a single 5 ns pulse in chromaffin cells clamped near the resting membrane potential (–70 mV) in these cells [5], which has also been detected for longer NEPs in GH3 and NG108 cells [7, 8] and also in bovine chromaffin cells [8]. A similar inwardly rectifying conductance was also detected in response

to twin NEPs in this study. Interestingly, the time course of onset (ms) and decay (min) of $I_{leak}$ elicited by a single 5 ns pulse [5], which may be the product of ion flux through lipid nanopores and/or voltage-independent ion channel proteins, correlated well with that of the rapid inhibition (ms to s) and delayed recovery (min) of $I_{Na}$ by twin 5 ns pulses observed in this study.

Because many types of ion channels are localized and regulated in highly organized lipid microenvironments such as lipid rafts and caveolae [43–45], another way that twin NEPs could produce a transient suppression of the number of active channels is by influencing endocytosis and membrane protein recycling, resulting in the transient disappearance of active $Na^+$ channels (discussed in the section below). Unequivocal evidence in support of this hypothesis will require further experiments in which the movement of fluorescently tagged voltage-gated $Na^+$ channels expressed in a mammalian cell line is monitored over time before, during and after the application of a single or twin NEPs by total internal reflection fluorescence or super-resolution fluorescence microscopy.

Another viable hypothesis to explain the inhibitory effects of NEPs on $I_{Na}$ is that NEPs may directly influence the $Na^+$ channel protein. Recent molecular dynamics simulations by Rems *et al*. [46] showed that pulsed electric fields could create alternate conductive pores near the voltage-sensitive domains (VSDs) of voltage-gated ion channels including $Na^+$ channels. These alternate pathways were shown to spontaneously evolve into more complex pores stabilized by lipid head groups and were accompanied by unfolding of the VSD from the channel. Unfolded VSDs were unable to regulate the opening/closure of the channel pore leading to a dysfunctional ion channel that could no longer respond to changes in transmembrane potential, which is in agreement with our electrophysiological measurements showing a decrease in maximal $Na^+$ channel conductance. These complex alternate conduction pathways, and possibly the ion channel pore itself (e.g. if the channel was locked into a subconductance state), could also possibly contribute to the induction of the so-called "leak" current, which would be interpreted as current flowing through these alternate ion channel pathways, lipid nanopores or other permeable defects caused by the pulsed electric field. However, such profound defects would have to be reversible to account for the fact that the maximal conductance of $Na^+$ channels fully recovered 9 min after the second NEP at 8 MV/m, especially at short twin pulse intervals.

## Paradoxical recovery of $I_{Na}$ after twin NEP-mediated inhibition

A very interesting observation of the present study was the paradoxical finding that shortening the interval between twin pulses led to a progressive recovery of $I_{Na}$, which after 9 min exceeded the level reached following a single NEP, and in some cases the level seen in unexposed cells (e.g. for 10 ms interval at 5 and 8 MV/m). Even for twin NEPs at 10 MV/m, there were clear signs of a partial time-dependent recovery over a 9 min period of recording after delivery of the pulse pair, and minimal additive effects relative to a single NEP. As mentioned in the previous section, similar effects were observed whether $I_{Na}$ was recorded using physiological salt solutions, or $K^+$- and $Ca^{2+}$-free solutions containing $Cs^+$ and TEA on both sides of the membrane, indicating that twin NEPs targeted $I_{Na}$ and that the recorded effects did not include contamination by voltage-dependent $Ca^{2+}$ and $K^+$ currents. The progressive attenuation of the inhibition of $I_{Na}$ by twin NEPs by shortening the interval between pulses correlates well with the interval-dependent enhanced recovery.

In the last few decades, our understanding of the structure of membrane proteins has expanded considerably with the rapid development of new microscopy techniques with unprecedented spatial and temporal resolution, improvement in crystallization methods for

X-ray diffraction imaging, and the development of new computer algorithms to reconstruct images [47, 48]. Despite these developments, our understanding of the properties of trans-membrane proteins, and in particular ion channels in their native environment, still remains limited because the vast majority of protein structures that have been uncovered were determined without the stabilizing effects of a surrounding lipid bilayer. Many recent studies have highlighted important roles for the lipid bilayer in dynamically regulating ion channel activity [6, 49–54]. Cholesterol is an integral lipid component that conveys many structural properties to the cell membrane as well as regulating the activity of ion channels in the lipid bilayer. This natural steroid is thought to influence ion channels by several mechanisms that include: 1) a direct interaction with the channel protein, 2) changes in the fluidity of the bilayer, which indirectly affect ion channel gating and conformational changes, and 3) compartmentalization of ion channels into spatially restricted lipid rafts [55–60]. Many classes of voltage-gated ion channels have been reported to be influenced by cholesterol, including voltage-dependent $Ca^{2+}$ and $K^+$ channels [56]. Also, many ion channels are found in caveolae, a subset of lipid rafts [60–63], which are organized for optimal compartmentalized signal transduction and turnover of membrane proteins. Importantly, cholesterol may also regulate phosphatidylinositol 4,5-bisphosphate ($PIP_2$) levels, which is a key second messenger molecule regulating multiple types of ion channels [64, 65]. Depletion of membrane cholesterol affected the cellular response to 600 nanosecond electric pulses [66]; the same pulses were also reported to initiate hydrolysis or depletion of $PIP_2$ in the plasma membrane [67, 68], which could, at least in part, explain the NEP-induced inhibition of voltage-gated channels [6–8]. Therefore, transient disruption of the phospholipid bilayer by a NEP could be a possible step leading to the subsequent inhibition of voltage-gated $Na^+$ channels and/or the slow recovery of $I_{Na}$ following the application of a pulse pair.

In this study, the effect of disrupting the structure of the phospholipid bilayer on $I_{Na}$ was first investigated in chromaffin cells unexposed to NEPs. Cell exposure to MβCD, an agent used in our experiments as a cholesterol depleting agent [69], produced a partial inhibition of the inward $Na^+$ current (Fig 9). Whether this effect was the result of a direct acute blocking effect of MβCD on $Na^+$ channels, cholesterol depletion *per se* and/or lipid raft disruption is unclear as time-dependent effects of the agent on $I_{Na}$ were not investigated. However, distinguishing between these alternative possibilities would be difficult to interpret because in one study, a 5 min preincubation was sufficient to deplete membrane cholesterol, disrupt lipid rafts and increase $Ca^{2+}$-activated $Cl^-$ currents in vascular smooth muscle cells [63]. In most studies, a preincubation with this agent for 15 min or more (30 min in our study), is a standard procedure to deplete membrane cholesterol and eliminate caveolae in cells in which these membrane structures are found [19, 56, 69–72]. Interestingly, in the presence of MβCD, a pulse pair with a 10 ms interval still inhibited $I_{Na}$ but its recovery was nearly abolished. In fact, the inhibition was accentuated and was similar to the near doubling of the inhibition produced by the pulse pair at longer intervals (e.g. $\geq 1$ s). These results suggest that the inhibition and recovery of $I_{Na}$ by twin pulses reflect different processes as they were completely dissociated by MβCD. At longer intervals, twin pulses led to cumulative functional disruption of $Na^+$ channels, perhaps by inflicting damage to the channel protein as suggested by the molecular dynamics study by Rems *et al.* [46]. At shorter intervals, we propose that the rapid timing of delivery of the second pulse stimulated the trafficking of newly synthesized and readily assembled $Na_V$ subunits from the Golgi and/or endoplasmic reticulum systems toward the membrane. This is supported by the observation that twin pulses applied at intervals shorter than 1 s led to a recovery of $I_{Na}$ that partially masked its inhibition by twin pulses, and exceeded the amplitude of $I_{Na}$ in control cells unexposed to NEPs, suggesting the recruitment of additional $Na_V$ subunits leading to a larger number of functional $Na^+$ channels in the membrane.

Consistent with this hypothesis was the fact that the enhancement of $I_{Na}$ by twin NEPs was not associated with a change in the voltage-dependence of activation and inactivation. Clearly, additional experiments will be required to investigate this mechanism in more detail.

## Conclusion

The results of this study have shown that $I_{Na}$ in excitable adrenal chromaffin cells can be differentially modulated by a single versus a pair of 5 ns pulses, with the differential effect of the pulse pair depending on both the time interval between the two pulses and the E-field amplitude. Of particular significance is that a pulse pair at 5 MV/m having a very short interpulse interval (10 ms) can not only attenuate the inhibitory effect of a single pulse on $I_{Na}$ but also lead to a paradoxical progressive recovery of the inward current. The enhanced recovery of $I_{Na}$ disappeared at longer interpulse intervals, an effect accentuated at a higher E-field. While the biophysical mechanisms underlying these complex effects on $I_{Na}$ are unknown and remain to be explored, our results nevertheless point to the potential of utilizing NEPs to modulate $Na_V1.7$ channel activity and hence cell excitability for therapeutic applications. As an example, the hyperactivity of $Na_V1.7$ channels in neurons that has been associated with producing pain could be transiently reduced by applying twin NEPs [73–75]. It is also possible that other classes of $Na_V$ subunits might be similarly or differentially influenced by these pulses, which could affect the excitability of other types of neurons in both the central and peripheral nervous systems. To address this possibility, future experiments will be to investigate the effects of NEPs on $I_{Na}$ in mouse adrenal chromaffin cells that is mediated by both $Na_V1.3$ and $Na_V1.7$ channels [76].

## Supporting information

**S1 Fig. Sample traces illustrating the timing of application of a single or pulse pair at E-field of 8 MV/m and recording of the inward Na current evoked by a constant voltage step protocol.** Traces in all panels were from different experiments. The constant voltage step protocol consisted of applying a 100 ms voltage clamp step to +10 mV from a holding potential as shown at the top of panel A. Traces of inward currents were continuously recorded by applying a total of 200 voltage clamp steps, with a 3 s interval between each step. The inset in panel A shows an expanded view of the peak inward current. Representative traces of inward current following exposure of a cell to a single 5 ns pulse (A) or a pulse pair with time interval of 10 ms (B), 100 ms (C), 1 s (D), 5 s (E) or 30 s (F). For panels A thru D, the single pulse or pulse pair (red line) was applied just prior to the 21st voltage clamp step in the sequence to record to current, with an interval of 1.5 s for the single pulse (P) or the first pulse (1st P) of a pulse pair. For the 10 ms (B) 100 ms (C) and 1 s data set (D), the second pulse (2nd P) of a pulse pair interval was applied with an interval of ~ 1.49, ~ 1.4 and ~ 0.5 s, respectively, prior to the application of the voltage clamp step. For panels E and F, because the interval was longer than the membrane recording sweep duration, the membrane current traces from different sweeps were superimposed and displayed by different colors. For the 5 s data set (E), the second pulse (black line) was delivered after the 22nd voltage clamp step with an ~ 2.5 s interval between the second pulse and recording the sodium current recording during the 23rd voltage clamp step (not shown). For the 30 s data set (F), the second pulse interval (black line) was ~ 1.5 s prior to recording the inward current during the 31st voltage clamp step.
(TIF)

**S2 Fig. Typical effects on $I_{Na}$ of twin NEPs applied at different time intervals and E-fields.** Traces in all panels were from different experiments. $I_{Na}$ traces were elicited by voltage clamp

steps to +10 mV from a holding potential of -70 mV as described in S1 Fig. Representative traces of $I_{Na}$ recorded before, and after the first (1st P) and second pulses (2nd P) at an E-field of 5 MV/m (A), 8 MV/m (C) or 10 MV/m (E), respectively. Representative traces of $I_{Na}$ recorded before, immediately after the twin pulses with an interval of 10 ms (After pulses) and 9 min after the application of the twin pulses (9 min after pulses) at an E-field of 5 MV/m (B), 8 MV/m (D) or 10 MV/m (F), respectively.
(TIF)

## Acknowledgments

The authors would like to thank Dr. Tarique Bagalkot for his assistance with cell preparations and Robert Terhune for electrical engineering support. The authors would also like to thank Wolf Pack Meats in Reno, NV for providing fresh bovine adrenal glands.

## Author Contributions

**Conceptualization:** Lisha Yang, Indira Chatterjee, Gale L. Craviso, Normand Leblanc.

**Data curation:** Lisha Yang, Sophia Pierce.

**Formal analysis:** Lisha Yang, Sophia Pierce.

**Funding acquisition:** Gale L. Craviso, Normand Leblanc.

**Project administration:** Gale L. Craviso.

**Supervision:** Gale L. Craviso, Normand Leblanc.

**Writing – original draft:** Lisha Yang, Normand Leblanc.

**Writing – review & editing:** Sophia Pierce, Indira Chatterjee, Gale L. Craviso.

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
