## [Decision Letter · Decision Letter 0]

20 Nov 2019

PONE-D-19-29052

Paradoxical effects on voltage-gated Na+ conductance in adrenal chromaffin cells by twin vs single high intensity nanosecond electric pulses

PLOS ONE

Dear Dr. Leblanc,

Thank you for submitting your manuscript to PLOS ONE. After careful consideration, we feel that it has merit but does not fully meet PLOS ONE’s publication criteria as it currently stands. Therefore, we invite you to submit a revised version of the manuscript that addresses the points raised during the review process.

The reviewers felt your work is of interest and the experiments are generally well-performed.  However, several comments were raised regarding the methodology and interpretation of the data.  Specific comments are noted below and should be addressed for each reviewer.  These include concern that it is unclear if the sodium channels are the primary target for the effect of the NEPs, or if alterations to other currents (calcium or potassium) might account for the observations.  The reviewers also request more representation of raw data, including current traces, to enable the reader to interpret the results for themselves.  Clarification regarding methodology and statistical comparisons are also required.

We would appreciate receiving your revised manuscript by Jan 04 2020 11:59PM. To enhance the reproducibility of your results, we recommend that if applicable you deposit your laboratory protocols in protocols.io, where a protocol can be assigned its own identifier (DOI) such that it can be cited independently in the future. For instructions see: http://journals.plos.org/plosone/s/submission-guidelines#loc-laboratory-protocols

We look forward to receiving your revised manuscript.

Kind regards,

Kevin P.M. Currie, PhD

Academic Editor

PLOS ONE

Journal Requirements:

2. In your data availability statement you write, "All relevant data are within the paper and its Supporting Information files." Please ensure you have provided the individual data points used to create the figures and determine means, medians and variance measures presented in the results, tables and figures (http://journals.plos.org/plosone/s/data-availability#loc-faqs-for-data-policy). If these data cannot be publicly deposited or included in the supporting information, e.g. due to patient privacy or ownership by a third party, explain why and explain how researchers may access them.

Reviewers' comments:

Reviewer's Responses to Questions

**Comments to the Author**

1. Is the manuscript technically sound, and do the data support the conclusions?

Reviewer #1: No

Reviewer #2: Yes

Reviewer #3: Partly

2. Has the statistical analysis been performed appropriately and rigorously? 

Reviewer #1: No

Reviewer #2: Yes

Reviewer #3: No

3. Have the authors made all data underlying the findings in their manuscript fully available?

Reviewer #1: Yes

Reviewer #2: No

Reviewer #3: Yes

4. Is the manuscript presented in an intelligible fashion and written in standard English?

Reviewer #1: Yes

Reviewer #2: Yes

Reviewer #3: Yes

5. Review Comments to the Author

Reviewer #1: PONE-D-19-29052

This manuscript presents results from experiments to examine the effect of two closely spaced high intensity nanosecond electric pulses on voltage-dependent sodium (NaV) current in bovine chromaffin cells. The twin pulses are varied by intensity and interval and clear differences are observed as these parameters are altered. While the effects on the inward current are interesting, there are no definitive experiments that point to a clear mechanism. In addition, there is one result that appears to suggest that NaV channels are not directly affected by these pulses, which is against one of the main conclusions of this work.

One of the first observations presented is that the NaV current ‘runs down’ with time. The authors speculate that this results from a slow leftward drift in the voltage-dependent gating parameters, which has been observed previously. However, the data presented in their earlier paper (on a single high intensity pulse) argues against such a shift. Figure 2a of that paper shows that peak inward current remains at +10 mV for the entire 15 min recording period. On the other hand, outward currents show a large reduction over the same time period at +50 mV. The slow loss of the ‘N’ shape of the outward current IV suggests that the current actually running down is voltage-dependent calcium (CaV) current, which is well known to express this phenomenon. Since the NaV current has not been isolated in these experiments, it seems that CaV current rundown is a likely explanation for the slow loss of peak inward current. If this is true, one wonders if these experiments are focused on the wrong current.

One of the most difficult aspects of this work to understand is why isolated NaV current was not studied. This was also the case in the previous publication. The authors argue that peak inward current is primarily comprised of NaV current, but at +10 mV it is clear that CaV, KV and Ca-activated K currents also contribute. If the effect of the nanopulses were larger, the author’s argument for a NaV channel effect would be strengthened. However, the reduction varies between 5-14%, which supports the idea that any of the above-mentioned currents could be mediating these effects. The experiments on isolated NaV currents should be done.

While the majority of data shown are time courses of inward current measured at +10 mV, Figure 6 shows both normalized and relative conductance vs. voltage (GV) curves. Figure 6C shows the relative GV curve, which appears to discount the hypothesis that this is an NaV channel effect. The relative conductance is only reduced at voltages > 0 mV by the twin nanopulses. If the nanopulse effect were due to a reduction in the number of active NaV channels (as proposed by the authors), one would observe a 14% reduction over the entire GV curve, not only at voltages > 0 mV. One possible explanation is that KV current is enhanced by the nanopulse. This idea is supported by the voltage-dependent activation of KV current shown in the previous publication (activation > -10 mV). If this were true, a shift in the reversal potential would likely be observed. Unfortunately, IV curves are not shown, so it not possible to see if such a shift exists. Another problem is that none of the discussed mechanisms can explain the preferential reduction of inward current at voltages > 0 mV. The relative GV curve shown in the previous publication shows a similar preferential effect at depolarized voltages.

In the Discussion section, the authors describe the possibility that the nanopulses produce new pathways for ions to permeate the cells as leak currents (not voltage-dependent). However, do data are presented on the holding current. If such ‘leak’ pathways were generated, there should be an increase in holding current if those pathways were Na+ or Ca2+ selective, or if they were non-selective for cations. If such leak pathways were K+ selective, holding current may not be significantly altered at -70 mV (the holding potential). Such an increase in K+ leak could be observed by making brief, strong hyperpolarizing steps (-120 mV perhaps) to look for increased holding current at a voltage with a driving force strong enough to observe K+ flux.

One thing missing from the figures is the actual ionic currents. The currents at +10 mV should be shown with the time courses for the three times used for the GV curves, namely control, peak reduction of inward current and at 9 min. It is also important that a biophysical analysis be done on the currents. At least the rise time of the inward current and information on the outward current(s) needs to be added. Otherwise the reader is not able to independently assess the author’s conclusions.

The authors state that the reduction of current due to the nanopulses is instantaneous, but does not appear to be true. There is a clear time course to the current reduction, which is easily observed for the larger nanopulse effects. Such a time course should be investigated as it is possible that clues to the mechanism could be gleaned from such an analysis.

The methyl-β-cyclodextrin experiment is interesting, but impossible to interpret without more information. The currents need to be shown so that the reader can see if outward currents are also affected. Here the authors also admit that there is a role for KV currents in shaping the inward current, since they hypothesize that a shift in the reversal potential is due to the increased influence of these currents. Even though the reversal potential is measured at a more depolarized voltage (~+40 mV) than +10 mV, this highlights the problems with interpretation of voltage clamp data from currents that are not isolated.

Reviewer #2: This research article addresses the effects of twin nanosecond electric pulses (NEP) on sodium ion (Na+) conductance (INa) in cultured adrenal chromaffin cells. Whole, single bovine cells isolated from a fresh adrenal medulla are voltage clamped and then exposed to NEP. The inhibition and recovery dynamics of INa in cells were compared for exposure to no pulse (control), single, and twin pairs of 5 ns pulses. Variation of electric field amplitude (5 MV/m, 8 MV/m, or 10 MV/m) and time interval between NEP pairs (10 ms, 100 ms, 1 s, 5 s, or 30 s) produced differential inhibition and recovery effects. All NEP exposures inhibited INa before recovery to some extent. Twin NEP exposures increased percent inhibition of maximal conductance over comparable single NEP exposures (except 5 MV/m and 10 ms interval), while twin NEP exposures with lower amplitudes and shorter intervals enhanced recovery. The mechanism underlying twin NEP inhibition and recovery of INa does not depend on a shift in voltage-dependence of steady-state activation and inactivation, suggesting that NEP exposure does not interfere with voltage sensors. Nor does NEP exposure change the reversal potential of inward current, suggesting ion selectivity and perhaps the structure of ion conduction pathways are maintained. Since maximal Na+ conductance was inhibited, the hypothesis that twin NEP exposure reduces the number of active channels is adopted to explain the mechanism of inhibition. As an initial experiment toward addressing this mechanistic hypothesis, lipid rafts are disrupted by universally depleting cholesterol within cells before exposure to NEP. Although there was minimal effect of cholesterol depletion on the response of cells to single NEP, it enhanced inhibition and nearly eliminated recovery of INa following twin NEP exposure. The important conclusion is that tuning of 5 ns electric pulse exposures can be used to modulate sodium ion channel activity in cells. The impact of this conclusion and the overall study make this well-written article appropriate for publication in PLOS ONE.

Minor revisions are recommended in order to address the following comments.

- All data does not appear fully available. Only mean values are presented within the manuscript and figures. Please make full data available either as supporting information or in a public repository.

- No primary experimental evidence linking caveolae to the results is presented, and methyl-b-cyclodextrin universally depletes cholesterol within cells. Cholesterol impacts numerous membrane properties and functions in addition to fusion, trafficking and recycling, such as bilayer fluidity and the second messenger molecule, as mentioned in the Discussion. Therefore, it is recommended to reword line 368 (perhaps “investigate” instead of “determine”), and to remove “capable of disrupting caveolae integrity” in line 506.

- Regarding the influence of cholesterol depletion on cellular responses to 10 ns and 600 ns electric pulses, consider citing this 2016 publication by Cantu et al. (https://doi.org/10.1016/j.bbamem.2016.07.006).

- Capitalization of the title is inconsistent with the others in Reference 47 (lines 710-713).

Reviewer #3: The manuscript by Yang et al., describes the effects of dual high intensity pulses on voltage-gated sodium channel currents in adrenal chromaffin cells. There are interesting findings on the recovery of the sodium channel current in response to dual rather than single NEPs and the effects of depletion of membrane cholesterol. The manuscript is generally well written but requires clarification on the methodology and statistical tests used which could determine whether some conclusions are in agreement with the data presented.

1) Throughout the manuscript, the authors conclude that there are differences in the magnitude of inhibition and / or recovery of voltage-gated Na+ channel currents in chromaffin cells exposed to dual NEPs in comparison to those exposed to a single NEP. However, with the exception of the data presented in figure 6, there is no mention of which statistical tests were used to conclude that the responses were significantly different. Could the authors provide the results from the statistical tests in the manuscript text and figures?

2) The authors concluded that there was an additive effect on inhibition only when the interval between the NEPs exceeded 100ms. With regards to this, could the authors clarify a couple of points in the methodology? In the methods, the authors state that a single NEP or pair of NEPs was applied between the 20th and 21st voltage steps with an interval of 0.5s between the time the NEP was delivered and recording of currents.

a) Did the voltage step depolarizations to +10mV every 3s continue during the interval between the two NEPs? In figures 1B, 3B and 4B (all 30s NEP intervals) there look to be data points that indicate the step depolarizations were continued in between exposure to the two NEPs.

b) Does the 0.5s interval between the time the NEP was delivered refer to the first NEP? If there is an interval of 0.5s between the first NEP of the pair and the subsequent step depolarization to record INa, is the second NEP exposure after the 21st voltage step for any cells in which the interval between two NEPs is 1s or greater? In addition, is the time between exposure to the 2nd NEP and the next step depolarization to record INa equal or different for each dual NEP exposure interval? For example, in the 10ms data is the 2nd NEP-voltage step interval ~ 490ms and in the 1s data the same interval ~ 2.5s?

It would aid the reader if the authors could depict this protocol, indicating after which voltage step the second NEP exposure occurred and the time interval between NEP exposure and the next step depolarization.

c) If the 2nd NEP-voltage step time interval is unequal, does varying the interval between a single NEP, of different electric field strengths, and a voltage step to record INa affect the voltage-gated sodium channel inhibition observed?

Minor comments:

1) In figures 1-4 is the same control or 1 pulse data set used in all panels? Could the authors state in the figure legends if this is the case.

2) Legend for Figure 1 – I think page 10 line 209 should read voltage step to +10mV rather than -10mV.

3) It would be useful to the reader to see some representative sodium current traces.

4) Could the authors provide more detail in the methods on the double pulse protocol used to determine steady state inactivation including conditioning pulse duration, test pulse duration and voltage step, interval between pulses. It would also be useful to the reader to provide a schematic of this protocol and the steady state activation protocol alongside the relevant data sets.

5) PLOS does not permit references to data shown. In the instance where this refers to the reversal potential, it may be helpful for the readers to see the IV curves that were used to determine steady state activation and conductance rather than remove the reference to this data.

6. PLOS authors have the option to publish the peer review history of their article (what does this mean?). If published, this will include your full peer review and any attached files.

Reviewer #1: No

Reviewer #2: No

Reviewer #3: No

---

## [Author Response · Author response to Decision Letter 0]

2 Apr 2020

All comments were addressed in the point by point response to all three reviewers.

---

## [Decision Letter · Decision Letter 1]

7 May 2020

PONE-D-19-29052R1

Paradoxical effects on voltage-gated Na+ conductance in adrenal chromaffin cells by twin vs single high intensity nanosecond electric pulses

PLOS ONE

Dear Dr. Leblanc,

Thank you for submitting your manuscript to PLOS ONE. After careful consideration, we feel that it has merit but does not fully meet PLOS ONE’s publication criteria as it currently stands. Therefore, we invite you to submit a revised version of the manuscript that addresses the points raised during the review process.

Reviewers 2 and 3 were unable to provide additional comments in the timeframe provided but in looking through your responses I found them to satisfy those previous critiques.  Reviewer 1 notes that you have done an excellent job revising the manuscript and has a few remaining comments that would be helpful to address.  These are mostly minor issues involving clarification and perhaps rewording of the discussion. The reviewer does suggest an experiment to determine if acute methyl beta cyclodextrin alters the sodium channel current, suggesting this is important when implicating lipid rafts in the altered response to the nanosecond pulses.  This would be useful data to provide.  However, if you are unable to do so in a reasonable time frame due to current work restrictions then please consider alternative approaches to addressing this concern.  For example, is there evidence in the literature, or can the conclusion be tempered / reworded to reflect the possibility that acute effects of the drug on the channel could alter interpretation.

We would appreciate receiving your revised manuscript by Jun 21 2020 11:59PM. To enhance the reproducibility of your results, we recommend that if applicable you deposit your laboratory protocols in protocols.io, where a protocol can be assigned its own identifier (DOI) such that it can be cited independently in the future. For instructions see: http://journals.plos.org/plosone/s/submission-guidelines#loc-laboratory-protocols

We look forward to receiving your revised manuscript.

Kind regards,

Kevin P.M. Currie, PhD

Academic Editor

PLOS ONE

Reviewers' comments:

Reviewer's Responses to Questions

**Comments to the Author**

1. If the authors have adequately addressed your comments raised in a previous round of review and you feel that this manuscript is now acceptable for publication, you may indicate that here to bypass the “Comments to the Author” section, enter your conflict of interest statement in the “Confidential to Editor” section, and submit your "Accept" recommendation.

Reviewer #1: All comments have been addressed

2. Is the manuscript technically sound, and do the data support the conclusions?

Reviewer #1: Partly

3. Has the statistical analysis been performed appropriately and rigorously? 

Reviewer #1: Yes

4. Have the authors made all data underlying the findings in their manuscript fully available?

Reviewer #1: Yes

5. Is the manuscript presented in an intelligible fashion and written in standard English?

Reviewer #1: Yes

6. Review Comments to the Author

Reviewer #1: PONE-D-19_29052_R1

The authors have done an excellent job responding to the previous criticisms. The new experiments now make it clear that NaV channels are affected by the high intensity electric pulses (NEP). The other major issues have also been resolved.

The double NEPs present several different effects over a single NEP. First, the size of the NaV current reduction is generally larger, as one would expect. Second, there is a recovery of NaV current amplitude, but only if the two pulses are ≤ 100 ms apart. Interestingly, this recovery is actually an over-recovery, since the current amplitude is even larger than the current would be under control conditions, at least for twin NEPs 10 ms apart and < 10 MV/m. Indeed, the lack of full recovery with 10 MV/m NEPs at 10 ms interval suggests a different mechanism involved to prevent recovery. The data are very interesting. The problem is that the Discussion does clearly present these ideas. The speculated mechanisms involved in the initial reduction of current are presented, but the ‘paradoxical’ recovery is not. It would be helpful to the reader if the authors would re-organize the Discussion to talk about the NaV current inhibition by NEPs, including the possible mechanisms, and then the paradoxical recovery, again including possible mechanisms. Further, the enhancement of the current beyond that in control (at 9 min) needs a mechanistic discussion. The recovery is discussed, but that discussion comes before the presentation of the mechanisms for the NaV current inhibition, and therefore there is an absence of mechanisms for the recovery.

One experiment that needs to be added is the effect of acutely applied MβCD on the NaV current. The authors show the effect of a 30 min MβCD pretreatment and conclude that the associated effects are consistent with lipid raft involvement, but this conclusion is only valid if the acute application shows no or significantly smaller effects. It is generally agreed that the 30 min incubation period is required for the disruption of lipid rafts by MβCD.

Other comments

1. Figure 1. Why is there apparently no CaV current when NMDG was used to replace Na+? Is there a small inward current in TTX that could be attributed to CaV channels? Clarification would be helpful.

2. Figure 6 legend. For the panels showing recovery, the measurement was taken at 9 min after the NEPs, but that time is not mentioned in the figure legend. It is mentioned in the text but should be added to the legend. In this figure, there is no mention of significant differences. It is not clear if the any of the presented valued in each of the graphs are different from one another.

3. Line 367. It is stated that K-gluconate was replaced with aspartate, which is not true. The replacement compound was actually Cs-aspartate, at least that is what I could glean from the Methods section. This needs to be corrected.

4. The description of the experiments measuring the leak current need to be clarified. Were these experiments done in the same cells used for the NaV current measurements? It is also not clear why figure 8 presents discrete points for the leak measurement instead of a continuous line, as would be recorded from a ramp. If these points were measured from the ramp, it should be clear how the measurement was done. For example, were a range of values averaged for each point? If so, how wide was the voltage range covered?

5. The conclusion seems to focus solely on the single NEP, since it describes only the NaV inhibition. There needs to be discussion here of the recovery as well. In addition, there needs to be discussion of the potential selectivity of the NEPs, since the focus here is on pain management. It seems very likely that other NaV channels will also be affected, which would impact other sensory axons.

7. PLOS authors have the option to publish the peer review history of their article (what does this mean?). If published, this will include your full peer review and any attached files.

Reviewer #1: No

---

## [Author Response · Author response to Decision Letter 1]

15 May 2020

Those are specifically addressed in two separate documents (Cover Letter and Response to Reviewer #1).

---

## [Editor Report · Decision Letter 2]

20 May 2020

Paradoxical effects on voltage-gated Na+ conductance in adrenal chromaffin cells by twin vs single high intensity nanosecond electric pulses

PONE-D-19-29052R2

Dear Dr. Leblanc,

We are pleased to inform you that your manuscript has been judged scientifically suitable for publication and will be formally accepted for publication once it complies with all outstanding technical requirements.

With kind regards,

Kevin P.M. Currie, PhD

Academic Editor

PLOS ONE
---

## [Editor Report · Acceptance letter]

29 May 2020

PONE-D-19-29052R2 

Paradoxical effects on voltage-gated Na+ conductance in adrenal chromaffin cells by twin vs single high intensity nanosecond electric pulses 

Dear Dr. Leblanc:

I am pleased to inform you that your manuscript has been deemed suitable for publication in PLOS ONE. Congratulations! Your manuscript is now with our production department. 

With kind regards,

on behalf of

Dr. Kevin P.M. Currie 

Academic Editor

PLOS ONE